# lumpR 2.0.0: An R package facilitating landscape discretisation for hillslope-based hydrological models

Tobias Pilz, Till Francke, and Axel Bronstert

Institute of Earth and Environmental Science, University of Potsdam, Germany

*Correspondence to:* Tobias Pilz (tpilz_at_uni-potsdam.de)

**Abstract.** The characteristics of a landscape pose essential factors for hydrological processes. Therefore, an adequate representation of the landscape of a catchment in hydrological models is vital. However, many of such models exist differing, amongst others, in spatial concept and discretisation. The latter constitutes an essential pre-processing step, for which many different algorithms along with numerous software implementations exist. In that context, existing solutions are often model specific, commercial or depend on commercial back-end software, and allow only a limited or no workflow automation at all.

Consequently, a new package for the scientific software and scripting environment R, called *lumpR*, was developed. lumpR employs an algorithm for hillslope-based landscape discretisation directed to large-scale application via a hierarchical multi-scale approach. The package addresses existing limitations as it is free and open source, easily extendible to other hydrological models, and the workflow can be fully automated. Moreover, it is user-friendly as the direct coupling to a GIS allows immediate visual inspection and manual adjustment. Sufficient control is furthermore retained via parameter specification and the option to include expert knowledge. Conversely, completely automatic operation also allows extensive analysis of aspects related to landscape discretisation.

In a case study, the application of the package is presented. A sensitivity analysis of the most important discretisation parameters demonstrates its efficient workflow automation. Considering multiple streamflow metrics, the employed model proved reasonably robust to the discretisation parameters. However, parameters determining the sizes of subbasins and hillslopes proved to be more important than the others, including the number of representative hillslopes, the number of attributes employed for the lumping algorithm, and the number of sub-discretisations of the representative hillslopes.

## 1 Introduction

Hydrological simulation models are important tools for gaining process understanding, forecasting streamflow, supporting water managers, and climate and/or land-use change impact studies. However, the lack of a unified theory in catchment hydrology led to a large number of competing models (e.g., Weiler and Beven, 2015). These differ in terms of aim and scope, process representation and parameterisation, temporal resolution, and spatial discretisation. The latter includes the size and shape of units where the model's underlying mathematical equations are solved.

The most straightforward procedure in terms of data handling and computation is the discretisation of the landscape into quadratic grid cells of equal size, commonly referred to as *fully distributed* approach. This, however, suffers from several

drawbacks: high computational burden and large memory demands for small grid cells, while for large grid cells limitations in representing landscape variability and processes at scales smaller than given by the grid's resolution may occur. Another strategy is to treat a hydrological catchment as a single unit, known as *lumped* approach. While being easy to implement along with simple and fast computable conceptual process representations, this approach fails to adequately represent complex

landscapes and process interactions often leading to a deteriorated simulation of hydrological catchment behaviour (e.g., Yang et al., 2000). Several approaches exist addressing this scaling problem for which no generic solution has been found so far (Beven, 2006). Subgrid parameterisation is a common procedure, describing the variability of a response function lying below a model's spatial resolution by the estimation of spatial distribution functions (e.g., Beven, 1995; Bronstert and Bárdossy, 1999; Nijzink et al., 2016). A different approach is to introduce different spatial levels in a model as compromise between distributed

and lumped discretisation schemes. This is sometimes referred to as *semi-distributed* approach, often being the preferred choice in practical applications (e.g., Kumar et al., 2010; Euser et al., 2015). Although no clear definition exists, this family of discretisation typically consists of a hierarchical multi-scale scheme dividing a hydrological basin into several subbasins which in turn contain irregularly shaped computational elements being hydrologically uniform entities (e.g., Krysanova et al., 1998). Thus, the properties of both distributed and lumped modelling can be found, often extended by integrating subgrid

parameterisation schemes. On the one hand, hydrological processes are simulated at different locations in the study area taking into account distributed model input (such as meteorological forcing and landscape parameters) and producing spatially variable output (such as lateral and vertical water flows). On the other hand, naturally heterogeneous but hydrologically similar areas are aggregated and parameterised in the same manner. The spatial heterogeneity of parameters or state variables of the model, such as hydraulic conductivity and soil moisture, respectively, may thereby be described by constitutive relationships.

In favour of more computational efficiency, the topology of individual elements is often neglected. This, however, imposes the threat that significant hydrological connectivity between the elements might not be correctly represented.

In the last decades, a large number of landscape discretisation procedures has been developed for the delineation of spatial units for hydrological models. The number of accompanying software solutions is even larger. This makes it a difficult task to choose a specific model, the corresponding discretisation approach, and potential tools for landscape pre-processing. There-

fore, the *first objective* of this paper is to provide an overview of existing landscape discretisation algorithms and software implementations, thereby solely focussing on semi-distributed hydrological model application (Sect. 2).

Among the semi-distributed approaches, hillslope-based modelling is an efficient option for representing heterogeneous runoff generation processes while accounting for phenomena of hydrological connectivity (see Sect. 2.2). However, so far only few computer programmes exist that aid in the pre-processing of hillslope-based models. Furthermore, these are often model-

specific, have a limited applicability, and are not freely available or can only be used along with commercial software (see Sect. 2.3). The *second objective* therefore is to present a new software package for the pre-processing of hillslope-based multi-scale hydrological models addressing these limitations. It is introduced and described in Sect. 3.

The role of detail of discretisation (i.e., the spatial resolution) of hydrological models, partly also as a user decision during the pre-processing process, has long been acknowledged in numerous studies (e.g., Wood et al., 1988; Kumar et al., 2010;

Han et al., 2014; Euser et al., 2015; Haghnegahdar et al., 2015; González et al., 2016). For grid-based models, this influence

has been thoroughly assessed (e.g., Zhang and Montgomery, 1994; Refsgaard, 1997; Molnár and Julien, 2000; Sulis et al., 2011). Such analyses are relatively straightforward, as changing grid resolution is easily attained. For semi-distributed models, however, systematic and objective analyses covering multiple scale are less common as they require an efficient and automated workflow for the creation of many realisations. The proposed software package is able to provide this. Thus, the *third and last* objective of this paper is to present an example case study in a semi-arid catchment using the WASA-SED model, thereby conducting a sensitivity analysis of crucial discretisation parameters (Sect. 4).

Eventually, the findings of this study are discussed and conclusions are inferred (Sects. 5 and 6, respectively).

## 2  Review of landscape representation in hydrological modelling

As being the starting point of any representation of landscapes in computer models, this section starts with a short overview of basic approaches for the representation of topography in computer models (Sect. 2.1). What follows is a review of methods for the delineation of spatial entities in semi-distributed hydrological modelling (Sect. 2.2) and, finally, an appraisal if and how these are supported by existing software tools (Sect. 2.3).

As a basis for watershed delineation and further landscape discretisation, a number of pre-processing steps have to be performed. These shall be briefly mentioned but not further discussed herein. Typically included is *(i)* pit-filling of the DEM to remove sink areas and ensure a proper drainage of water from the catchment; *(ii)* computation of flow directions; *(iii)* computing upslope contributing area (i.e., the accumulated flow for each DEM unit); *(iv)* derivation of the river network, typically based on flow accumulation from the previous step; *(v)* delineation of the hydrological basin and subbasins. For each step different algorithms have been proposed whose selection depends on the type of DEM, model to be employed, aim and scope of the study, or just personal preferences, e.g., in terms of software to be used (e.g., O'Callaghan and Mark, 1984; Tarboton et al., 1991; Costa-Cabral and Burges, 1994; Lacroix et al., 2002; Vivoni et al., 2004; Moretti and Orlandini, 2008).

### 2.1  Topography representation in computer models

Topography is typically mapped using Digital Elevation Models (DEMs). *Contour-based* DEMs store terrain information as contour lines (or x, y coordinate pairs) of specific elevation (Moore et al., 1988). Moore and Grayson (1991) and Maurer (1997) show example applications of contour-based terrain analysis for hydrological model application where this type of DEM proved to be powerful as its structure is based on the way how water flows on (albeit not necessarily below) the land surface. However, although they have been further investigated for hydrological application (e.g., Dawes and Short, 1994; Maunder, 1999; Zhang et al., 1999; Moretti and Orlandini, 2008) contour-based DEMs come along with some limitations. They have a relatively high data storage demand, topographic attributes are complicated to derive, and they provide no computational advantages (Moore et al., 1991).

*Triangulated Irregular Networks* (TINs) form a type of DEM sampling elevation points at specific landscape features, such as peaks or ridges, and form an irregular network of x, y, and z coordinates. They are very flexible as, due to their irregular structure, they are able to map regions of high heterogeneity with more data points than smooth terrain and thus avoid redun-

dancy and increase data storage efficiency (Moore et al., 1991; DeVantier and Feldman, 1993). TINs also proved to be useful in a number of hydrological applications (e.g., Tucker et al., 2001; Vivoni et al., 2004; Ivanov et al., 2004; Freitas et al., 2016). Their irregularity, however, makes the computation of topographic attributes more difficult and there can be problems when determining upslope connections for watershed derivation (Moore et al., 1991).

5   *Grid-based* DEMs store elevation information as a regularly spaced mesh. There are a number of drawbacks as the regular structure might impose artefacts and discontinuities while sub-resolution landscape features cannot be captured limiting the applicability for hydrological purposes. Furthermore, when increasing the grid's resolution to reduce these problems, computational burden and memory requirements are increased reducing their suitability for large scale applications. Nevertheless, grid-based DEMs are the most widely used data structures due to their straightforward generation from remote sensing data, 10   direct applicability for further investigations, and efficient calculation of geomorphological characteristics (Moore et al., 1991; DeVantier and Feldman, 1993).

Some models explicitly use TINs, such as tRIBS (Ivanov et al., 2004), or modelling units derived from contour-based DEMs, e.g., THALES (Grayson et al., 1992). The majority, however, delineate irregularly shaped polygons as computational model units derived from grid-based DEMs which will be further discussed in the following subsection.

15   **2.2   Discretisation approaches in semi-distributed hydrological modelling**

There exists a large number of landscape discretisation schemes for semi-distributed hydrological model application. In this context, with discretisation we understand the process of deriving computational units for a hydrological computer model from spatial input data. We do not consider temporal resolution here. The spatial discretisation in a model determines size, hierarchy and topology of model elements. For semi-distributed hydrological models, these comprise landscape elements such as 20   (sub-)catchments, river segments, hillslopes, hillslope segments (e.g., different slope sectors along a hillslope), hydrologically homogeneous areas, soil units and horizons. In correspondence to the dominating hydrological processes, the objects of higher hierarchy are commonly determined by topography, while for the lower hierarchy soil and vegetation are the distinguishing factor.

For our review we identified four general classes of discretisation approaches which are presented along with specific examples 25   in Tab. 1. These include *Areal unit* schemes which delineate spatial polygons as fundamental modelling units; approaches taking *Hillslopes* as spatial elements; *Functional units* with focus on a homogeneous process description rather than mere spatial entities; and a *Mixed* class, typically comprising a hierarchical scheme of different spatial scales. With the latter we mean conceptions exceeding the common *watershed > subbasin > element* scheme. In the following, the presented approaches shall be briefly described whereas more information on the mentioned software solutions can be found in Sect. 2.3.

30   Wood et al. (1988) focussed on how to define a hydrological (sub-)catchment and studied its dominant controls. They defined the smallest discernible averaging watershed where statistics of runoff generation did not further change as *Representative Elementary Area* (REA) and applied simple conceptual equations for the simulation of runoff generation. They found the size of their REA to be primarily influenced by topography. Similar, though conceptually different, is the study of Kouwen et al. (1993) who were looking for an approach to represent watershed heterogeneity over large basins. They defined *Grouped Re-*

**Table 1.** Classification of prominent landscape discretisation approaches for semi-distributed hydrological modelling. Approaches are ordered as they appear in the text. For the meaning of abbreviations of the approaches see text. Listed model and software solutions are non-exhaustive examples. Key references refer to the introductory or an illustrative example publication.

| Approach | Class | Model | Software solution | Key reference |
| --- | --- | --- | --- | --- |
| REA | Areal unit | – | – | Wood et al. (1988) |
| GRU | Areal unit | SIMPLE, CHARM | WATFLOOD | Kouwen et al. (1993) |
| ASA | Areal unit | SLURP | SLURPAZ | Kite (1995) |
| Hydrologic landscapes | Areal unit | – | – | Winter (2001) |
| HRU | Areal unit | PRMS, SWAT, SWIM, PREVAH, GSFLOW, MHYDAS | IOSWAT, AGWA, AVSWAT, WINHRU, Geo-MHYDAS | Leavesley et al. (1983); Flügel (1995) |
| HSB | Hillslope | h3D | – | Troch et al. (2003) |
| Single Hillslope | Hillslope | KINEROS, IHDM, CATFLOW, HILLSLOPE | – | Bronstert (1999) |
| Representative Hillslope | Hillslope | WEPP | GeoWEPP, LUMP | Flanagan and Nearing (1995) |
| ECS | Hillslope | – | SMART | Khan et al. (2014) |
| flow-interval scheme | Hillslope | – | – | Yang et al. (2002) |
| REW | Functional unit | – | – | Reggiani et al. (1998) |
| Functional response units | Functional unit | TAC | – | Uhlenbrook and Leibundgut (2002); Zehe et al. (2014) |
| Multiple scales | Mixed | RHESSys, WASA-SED | LUMP | Güntner (2002) |

*sponse Units* (GRUs), small watersheds as computational units with uniform meteorological forcing that are considered to be hydrologically heterogeneous by consisting of a range of land cover (or some other attribute's) characteristics where only percent cover is used for characterisation instead of an explicit spatial reference. Topography and the other relevant attribute (e.g., land cover) are assumed to be the major factors influencing runoff generation. A similar conception is utilised by the

5    *Aggregated Simulation Area* (ASA) approach developed for the SLURP model (Kite, 1995). Winter (2001) aimed for an integrated inspection of the complete hydrological system in different terrain types by introducing *hydrologic landscapes*. These consist of variations and multiples of fundamental hydrological landscape units as building blocks characterised by landsurface form, geologic framework, and climatic setting to describe movements of surface water, groundwater, and atmospheric water, respectively.

10    The concept of *Hydrological Response Units* (HRUs) is directly related to a smaller spatial level. Introduced by Leavesley et al. (1983) for their Precipitation-Runoff Modeling System (PRMS) and further elaborated by Flügel (1995), it evolved

to a prominent landscape discretisation scheme utilised by many models. An HRU is assumed to be a homogeneous set of hydrological process dynamics formed by a pedo-topo-geological association with specific land-use and as such controlled by land-use management and physical landscape properties. The conception has been adopted for models such as SWAT (Manguerra and Engel, 1998), SWIM (therein termed *hydrotopes*) (Krysanova et al., 1998), PREVAH (Viviroli et al., 2009),
or GSFLOW (Markstrom et al., 2008). MHYDAS is a process-based hydrological model for which the HRU concept has been further pursued for application in agricultural management contexts by including man-made hydrological discontinuities such as ditches and field boundaries (Moussa et al., 2002). However, the HRU approach commonly does not preserve topological information for the spatial units. Instead of a direct representation of water flow pathways, generated runoff is typically summed over all HRUs of a watershed and routed along a representative channel element.

Other approaches divide the watershed into *representative hillslopes* as one- or two-dimensional approximation of a three-dimensional soil catena separated by drainage network and ridges. For instance, Troch et al. (2003) developed the hillslope-storage Boussinesq (HSB) equation to simulate drainage and soil moisture storage dynamics along a hillslope described by a polynomial function. In their hybrid-3D hillslope hydrological model (h3D), Hazenberg et al. (2015) employ the latter along with the Richards' equation for vertical flow and a diffusive wave approximation of the shallow water equations for
overland flow as an efficient physically-based modelling approach aimed for use in continental and global scale Earth system models. Flanagan and Nearing (1995) introduced WEPP, a complex process-based soil erosion prediction model applicable over hillslopes or small watersheds comprised of multiple hillslopes, channels, and impoundments. They lump individual hillslopes by calculating and averaging quantitative hillslope characteristics. Examples for models treating single hillslopes over smaller scales include KINEROS (Woolhiser et al., 1990), IHDM (Beven et al., 1987), HILLSLOPE (Bronstert, 1994), or
CATFLOW (Maurer, 1997). Several studies focussed on how to delineate and describe hillslopes from a DEM (e.g., Cochrane and Flanagan, 2003; Noël et al., 2014), discussing morphometric controls on hillslope parameters (Bogaart and Troch, 2006), or investigating the role of hydrologic connectivity (Smith et al., 2013). Khan et al. (2014), for instance, formulated *Equivalent Cross Sections* (ECSs) as representatives of a part or an entire subbasin and proposed different averaging algorithms based on topographical and physiographical properties. Yang et al. (2002) introduced the *flow-interval hillslope scheme* where a
catchment is subdivided into a number of connected flow intervals which are defined by the width (i.e., the number of streams) and the geomorphological area (i.e., the drainage area) as functions of distance from the watershed outlet. Overall, hillslope hydrological modelling has been proven to be useful in regions with steeply sloping landscapes and heterogeneous runoff generation mechanisms where lateral water fluxes are relevant (Bronstert, 1999). On the other hand, especially when iterating through individual hillslopes of a watershed, applicability is clearly limited to smaller scales due to the large computational
demand.

In a different concept, a stronger focus is put on *functional units* rather than mere spatial units which always came along with the assumption of homogeneous process dynamics in a certain area. Herein, each unit is characterised by a specific dominant process and an accompanying model conceptualisation, such as for the TAC model of Uhlenbrook and Leibundgut (2002). Reggiani et al. (1998) delineated the basin into autonomous functional subbasins, the *Representative Elementary Wa-*
*tersheds* (REWs), using the drainage network as basic organising structure. The REW is then divided into five functional

sub-regions whereas micro-scale physical conservation equations for each sub-region are simplified and averaged, further accounting for thermodynamic exchange between sub-regions and REWs. Zehe et al. (2014) propose three *functional response units* separating radiation-driven vertical flow from rainfall-driven lateral flow processes on similar functional entities within a hydro-geomorphic homogeneous subbasin. However, it still is challenge how to define the size of averaging volumes and the
closure relationships of boundary fluxes (Beven, 2006).

Regardless of the chosen approach, for the application at large scales it is necessary to find a compromise between sufficient detail in landscape representation and computational feasibility. Thus, several studies examined the impact of discretisation complexity on model performance. They showed that (semi-)distributed models are usually more suitable for the representation of landscape heterogeneity and the exploration of hydrological processes, and are also more able to reproduce observed
discharge dynamics than lumped models (Kumar et al., 2010; Euser et al., 2015). However, there is a threshold of subdivision level above which no more improvements can be achieved (Wood et al., 1988; Han et al., 2014; Haghnegahdar et al., 2015). On the other hand, natural variability outside the models' spatial discretisation level can still exhibit an important limitation in the representation of natural processes. This can be accounted for by different spatial scales in a model application (e.g., the scales of meteorological forcing, model application, and basin characteristics) and combining them using parameter regionalisation
(e.g., Samaniego et al., 2010).

The concepts described above are often extended by a hierarchical multi-scale discretisation scheme. It commonly includes the basin of investigation being discretised into subbasins (i.e., hydrological sub-catchments) and hydrologically homogeneous modelling elements (e.g., HRUs). Thus, elementary units are grouped into a structure of higher order, summarising (and potentially defining a topology of) their in- and outputs. For the modelling system RHESSys, for instance, the landscape is
partitioned into a hierarchy of progressively finer units modelling different processes associated with a particular scale. A given spatial level is represented as object type with a set of states, fluxes, process representations, and corresponding model parameters (Band et al., 2000; Tague and Band, 2004).

For their WASA model, Güntner and Bronstert (2004) developed an even more complex scheme of six spatial levels. Starting at the watershed, subbasins are delineated, sub-divided into representative hillslopes termed landscape units which are further
separated into specific parts of the hillslope, the terrain components to account for lateral redistribution of water flows, whereas vertical processes and runoff generation are simulated over individual homogeneous soil-vegetation components for which a representative soil profile with respective soil layers has to be given. As for the GRU approach, at the smaller scales an explicit spatial representation is omitted in favour of a percentage cover representation in order to better capture landscape heterogeneity while keeping storage and computational demands at a minimum. This concept has been proven to be efficient
and successful for the simulation of heterogeneous semi-arid landscapes with complex hillslopes and patchy vegetation over large scales dominated by Hortonian overland flow and runoff redistribution mechanisms. As hydrologic connectivity of the landscape can be represented in a realistic manner, the model has been used for a number of studies investigating runoff redistribution and erosion processes (e.g., Güntner and Bronstert, 2004; Mueller et al., 2010; Medeiros et al., 2010; Bronstert et al., 2014).

## 2.3 Software for model pre-processing and landscape discretisation

The previous section described conceptual approaches of landscape discretisation and gave examples for models where these concepts are utilised. Their implementation into a model, however, requires a number of more or less complex pre-processing steps. Together with improving computer facilities and increasingly available DEMs and processing algorithms, software for terrain analysis and discretisation started to evolve. Already in the early 1990s, DeVantier and colleagues published a review of applications of *geographical information systems* (GIS) in hydrological modelling (DeVantier and Feldman, 1993). Especially for grid-based models, many tasks during spatial data pre-processing can be performed with standard GIS functionality. However, other steps require more specific operations. Thus, many researchers started writing their own scripts tailored to their needs and sometimes later on published or distributed their solutions both commercially and in non-profit manners.

TAPES-G is an early terrain analysis programme written in FORTRAN-77 and C for use on Unix machines already including several algorithms for specific tasks, e.g., five methods alone to calculate flow accumulation (Gallant and Wilson, 1996). A prominent and widely used example is the TOPAZ software package for automated analysis of digital landscape topography addressed to guide farmers, engineers, and scientists in both research and practical application. The programme is free of charge and available on request but the development stopped in 1999 (Garbrecht and Martz, 1999). Tarboton (2003) introduced TauDEM, a freely usable terrain analysis programme for Windows that can be applied independently from the command line or comes along with a *graphical user interface* (GUI) as extension for the commercial ArcMap software and is still being further developed. More information on hydrologically relevant software of the commercial ArcGIS family, such as ArcHydro, can be found at the software's community web pages: http://resources.arcgis.com/en/communities/hydro/ (accessed 23 June 2016). The *free and open-source software* (FOSS) GIS GRASS provides a number of elaborated and still evolving tools for hydrological model pre-processing, such as *r.watershed*, that have been successfully applied in a number of studies (e.g., Kinner et al., 2005; Metz et al., 2011; Neteler et al., 2012). Another recent example of stand-alone software is GeoNet, a tool for automatic channel head, channel network, and channel morphological parameter extraction from high resolution topography data that can be employed within MATLAB or, in a more recent version, as Python programme (Passalacqua et al., 2010; Sangireddy et al., 2016).

As being a frequently employed discretisation scheme used in well-known hydrological models such as SWAT, many software solutions exist implementing the HRU concept. Typically, the delineation process is based on the intersection of spatial raster data including land cover, soil, and/or geology. The programmes basically differ in terms of additional processing steps (such as terrain analysis), used algorithms, supported data formats and whether they are tailored to a specific model or are stand-alone, GIS back end (mostly ArcGIS or GRASS), supported operating systems, and whether they provide a GUI or have to be run from command line. Sanzana et al. (2013) developed a number of stand-alone and model independent Python scripts for terrain analysis and HRU mesh generation making use of GRASS functionalities. Also relying on GRASS and Python, Schwartze (2008) created an extension for QGIS (a FOSS GIS with user-friendly GUI) as HRU delineation tool. Other software is model specific such as IOSWAT (Haverkamp et al., 2005), AGWA (Miller et al., 2002, 2007), or AVSWAT (Di Luzio et al., 2004) being addressed to SWAT, WINHRU (Viviroli et al., 2009) written for PREVAH, or Geo-MHYDAS

(Lagacherie et al., 2010) which is a collection of SHELL and PERL scripts using GRASS to help users of MHYDAS with the model pre-processing. Some of these are mere wraps around individual programmes to guide through the whole modelling process, including terrain analysis, HRU delineation, preparation of input files, model execution and parameter calibration, and graphical and/or statistical analysis of simulation results.

The pool of software packages for other landscape discretisation schemes is less rich. Lacroix et al. (2002) presented SLURPAZ, an interface between the TOPAZ terrain analysis tool and the SLURP model for the delineation of ASAs. The WATFLOOD flow forecasting system is a framework consisting of a hydrological model (CHARM) including pre- and post-processor, incorporating the GRU approach (Kouwen, 2016) . Around the hillslope-based WEPP model, the geo-spatial assessment software GeoWEPP has been developed integrating TOPAZ, WEPP, and other tools for detailed analysis of spatially and

temporally variable environmental and management scenarios (Renschler, 2003). It is nowadays integrated into the ArcGIS project with a lightweight web-based interface for less advanced users and ad-hoc model application (Flanagan et al., 2013). RHESSys comes with interfaces for both GRASS and ArcGIS to assist in landscape pre-processing (Band et al., 2000). Ajami et al. (2016) published SMART, a MATLAB toolbox integrating TauDEM as terrain analysis tool, performing rainfall-runoff simulations over hillslopes in the sense of ECSs with several options for their derivation, and providing functions for the

post-processing of modelling results.

Francke et al. (2008) published an algorithm for the semi-automated delineation of representative hillslopes. This *Landscape Unit Mapping Program* (LUMP) first discretises the landscape into various hillslopes, the elementary hillslope areas, and computes a representative catena for each of them. As a next step, similar catenae are grouped into landscape units whereas for the classification several variables can be taken into account such as horizontal and vertical catena length, shape of the

profile, and sets of supplemental attributes further characterising the hillslope, e.g., qualitative data such as soil type and land cover class, or quantitative attributes such as LAI or further terrain characteristics. These representative catenae are eventually sub-divided into terrain components, e.g., into upslope, middle, and downslope parts. The approach will be further discussed in Sect. 3.2 and is illustrated in Fig. 2. LUMP is semi-automated in way that the hillslope-based landscape parameterisation is largely automated generating reproducible results and reducing required user decisions to a minimum whereas, on the other

hand, expert knowledge can be easily incorporated to improve the discretisation outcome. Contrary to other hillslope-based algorithms, due to the discretisation at multiple scales, it is applicable over large areas with relatively little effort. Even though LUMP was directed to the pre-processing of the WASA-SED model (Güntner and Bronstert, 2004), it is stand-alone and the output can as well be used for other hillslope-based models. However, the programme is basically a collection of freely available scripts written in MATLAB and SHELL using GRASS functionalities. Additionally to the dependence on commercial

software, the workflow still requires a considerable number of pre-processing steps and user interaction.

Considering the above-mentioned merits of hillslope-based landscape discretisation, the number of tools for automating this tasks is low. On the other hand, manual derivation of such an discretisation is labour intensive, prone to error and rarely fully reproducible, which generally precludes its application on the larger scale. Thus, to meet the second objective of this study to develop a user-friendly and efficient tool for hillslope-based landscape discretisation, it was decided to build upon the LUMP

algorithm which already remedies some of the above-mentioned shortcomings.

## 3 lumpR: R package description

The *landscape **u**nit **m**apping **p**rogram for **R*** (lumpR) was developed with the aim to obtain a lightweight user-friendly and efficient tool for hillslope-based landscape discretisation and serving as pre-processing tool for the WASA-SED model (Güntner and Bronstert, 2004, see also Sect. 4.2). In a more general sense, however, it should meet the requirements of being *(i)* platform independent, *(ii)* applicable for other hillslope-based models, too, *(iii)* free and open-source, *(iv)* automated as far as possible reducing subjectivity but *(v)* allowing to include expert knowledge. In order to produce an easily applicable software and to meet objectives (i) and (iii) in particular, it was decided to use the scripting language R (R Core Team, 2015) and assemble *lumpR* as a software package for this environment, licensed under the GNU General Public License (GPL) version 3 or later.

Figure 1 gives an overview over structure and functionalities of the package. In the following these shall be explained in more detail. For more information on how to install and use the package and to inform about updates the reader is referred to the package's documentation (see Sect. 7).

### 3.1 Prerequisites and general workflow

As it is a package of the scripting language R, lumpR requires the statistical software R together with various packages it depends on. It employs a number of external calls to GRASS GIS and thus requires having GRASS to be installed. A third requirement is a *database management system* (DBMS) which will be accessed via the *open database connectivity* (ODBC) which has to be set up as well. So far the DBMSs MySQL/MariaDB, SQLite, and MS Access are supported. When all settings and algorithm parameters can be determined, all processing steps can be run completely automatically. However, it is generally recommendable to process the steps successively to check the intermediate results, if necessary.

After installing the package and all additionally needed software, the user has to acquire and prepare all needed spatial data in a location in GRASS GIS. Internally the package's functions connect to that location, use the given data for processing while partly employing GRASS functions, and finally stores the spatial output in the GRASS location and/or text files in a specified directory for immediate inspection after each function call. As a first step in a new R session, before executing any of the package's function, R has to be connected to the GRASS location by the user. A template script guiding through the processing steps of landscape discretisation and the parameter database management has been prepared and is provided along with the package (see Sect. 7).

### 3.2 Landscape discretisation

As is sketched in Fig. 1, the process of landscape discretisation involves five functions that should be applied in the following order. This can be ensured by customising the provided template script. Figure 2 gives an illustrative example for the outcomes of the following steps (i) to (iv). *(i) calc_subbas()* sub-divides the hydrological basin into subbasins using a given grid-based DEM (black outlined polygons S1–3 in Fig. 2). Subbasin size can be influenced by the user by either giving a set of coordinates of drainage locations inferred beforehand or by specifying the parameter thresh_sub being the minimum size of a subbasin in number of grid cells internally used by GRASS function r.watershed. Furthermore, the river network is inferred

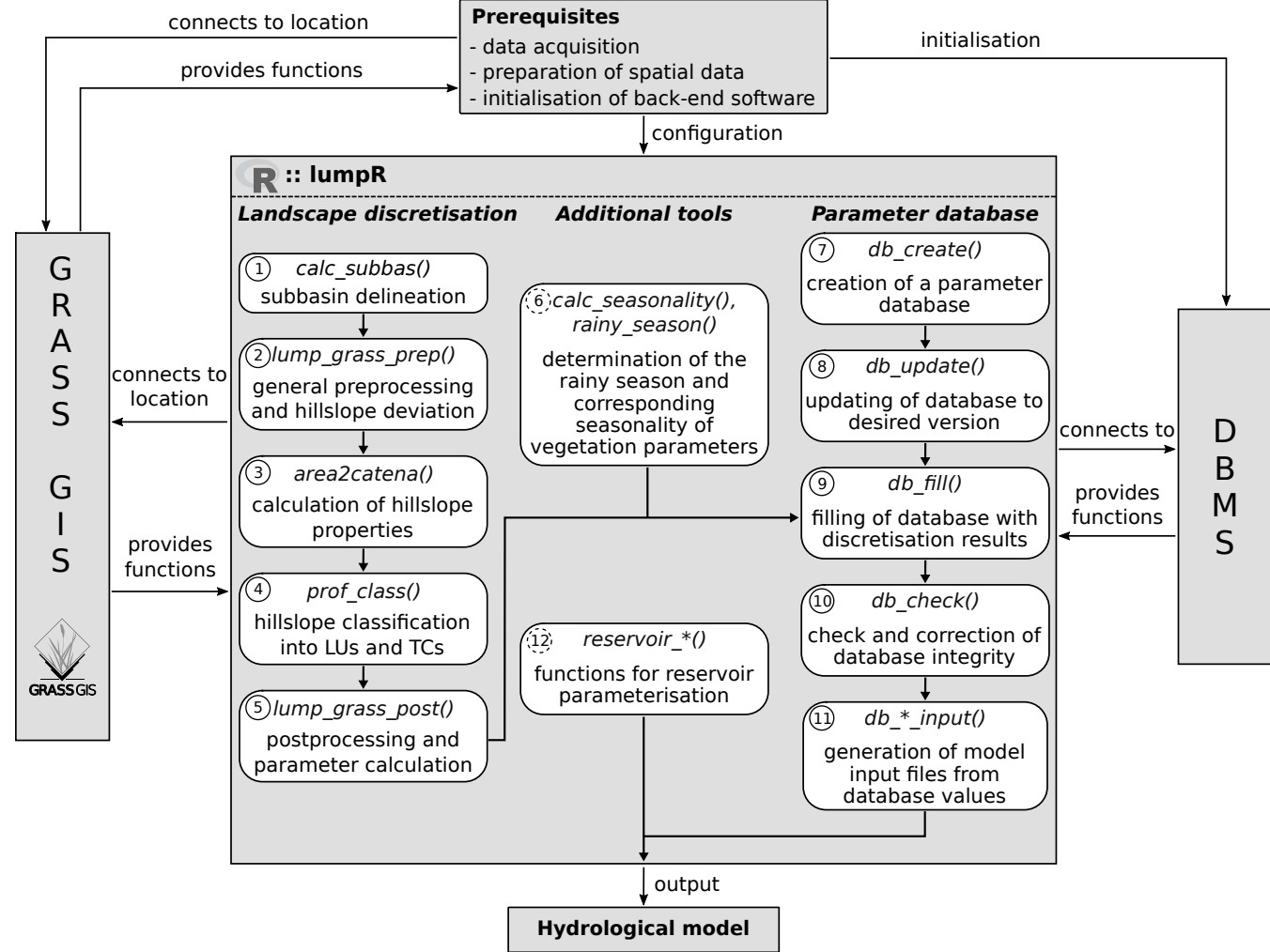

**Figure 1.** Schematic overview of the functionalities of the lumpR package. Shown are all available functions (*in italics*) with a short explanation, interactions with external components, and the typical flow of information during the application. Processing order is indicated by numbers, whereas optional steps are indicated by dashed circles. Note that the acronym DBMS refers to *DataBase Management System*.

from calculated flow accumulation (the number of upstream raster cells draining through a specific raster cell) via a user-defined threshold (blue lines in Fig. 2). *(ii) lump_grass_prep()* does several pre-processing steps needed for later use such as computing *Soil-Vegetation Components* (SVCs) as simple overlay of soil and vegetation raster maps, inferring Horton stream order, and DEM-processing steps including the calculation of flow direction, flow accumulation, relative elevation (i.e., elevations above next downstream river grid cell), and distance to next river grid cell. The results are stored in the specified GRASS location. Furthermore, the function infers *Elementary Hillslope Areas* (EHAs) based on the size parameter eha_thres. These are the

basic units for calculation of representative catenae and, thus, one can think of them as single hillslopes (denoted as small polygons in Fig. 2).

*(iii) area2catena()* takes data from step (ii) and supplemental raster maps of quantitative and/or qualitative attributes to calculate a representative catena for every EHA (gray boxes in Fig. 2). It is characterised by horizontal and vertical length, shape
(in terms of cumulated elevation along the hillslope), slope width (approximated by taking the number of grid cells at a profile point divided by the total number of grid cells representing the whole hillslope), and all supplemental data. This reduction is based on the work of Cochrane and Flanagan (2003) for the WEPP model. *(iv) prof_class()* classifies the representative catenae into *Landscape Units* (LUs) (coloured areas in Fig. 2). In this step similar catenae are identified and lumped together based on the calculated properties employing an unsupervised K-means clustering method. The user has to specify the number of classes
to generate from each attribute during the clustering, which is done separately for each attribute. The final class assignment for each catena results from the combination of these attribute-wise classifications. Each LU is then further sub-divided into *Terrain Components* (TCs), i.e., planar elements representing, e.g., upper, middle, and downslope parts of the lumped catena (coloured diagrams in Fig. 2). The number of TCs to be generated for each LU can be specified and the partition is done by evaluating the derived LU properties and employing a minimisation of variances approach. Topological relations between
SVCs, TCs, and LUs are established, expressed as percentages of covered area and along-slope location of TCs within a LU rather than spatial coordinates. For visual inspection, the user has the option to let the functions generate plots during steps (iii) and (iv). The employed algorithms for steps (iii) and (iv) are explained in more detail by Francke et al. (2008).

Finally, *(v) lump_grass_post()* establishes the topology between subbasins and LUs, again expressed as percentage of covered area. Subbasins are the only spatial units with explicit reference in terms of geographic coordinates. In addition, subbasin
and LU specific parameters such as representative channel geometry and routing parameters, groundwater, and landscape co-efficients, are approximated. Hereby, rather simple relationships or typical standard values are employed. The output of this function is mainly designed to provide a complete plausible parameterisation. Where alternative information are available, they should be used. See the function's documentation for more details.

A summary of the most important parameters for the landscape discretisation process is given in Tab. 2. Their meaning along
with a sensitivity analysis will be further discussed in Sect. 4.4.

### 3.3 Additional tools

In order to meet further capabilities of the WASA-SED model, functions *reservoir_*()* have been introduced. They facilitate the pre-processing of reservoir-specific input files for the model using spatial reservoir data and pre-compiled parameterisations. Additional function *rainy_season()* calculates start and end dates of the rainy season for every year based on a time series of
daily precipitation values using a statistical approach described by Gerstengarbe and Werner (1999). *calc_seasonality()* then uses the output of the former function and information about seasonal variation of a vegetation parameter to calculate a daily time series of that parameter by linear interpolation of the parameter's node points depending on the current start and end dates of the rainy season for a specific year. In hydrological models such as WASA-SED or WaSiM-ETH (Schulla and Jasper, 2007), such information can be used to describe intra-annual variations of vegetation parameters.

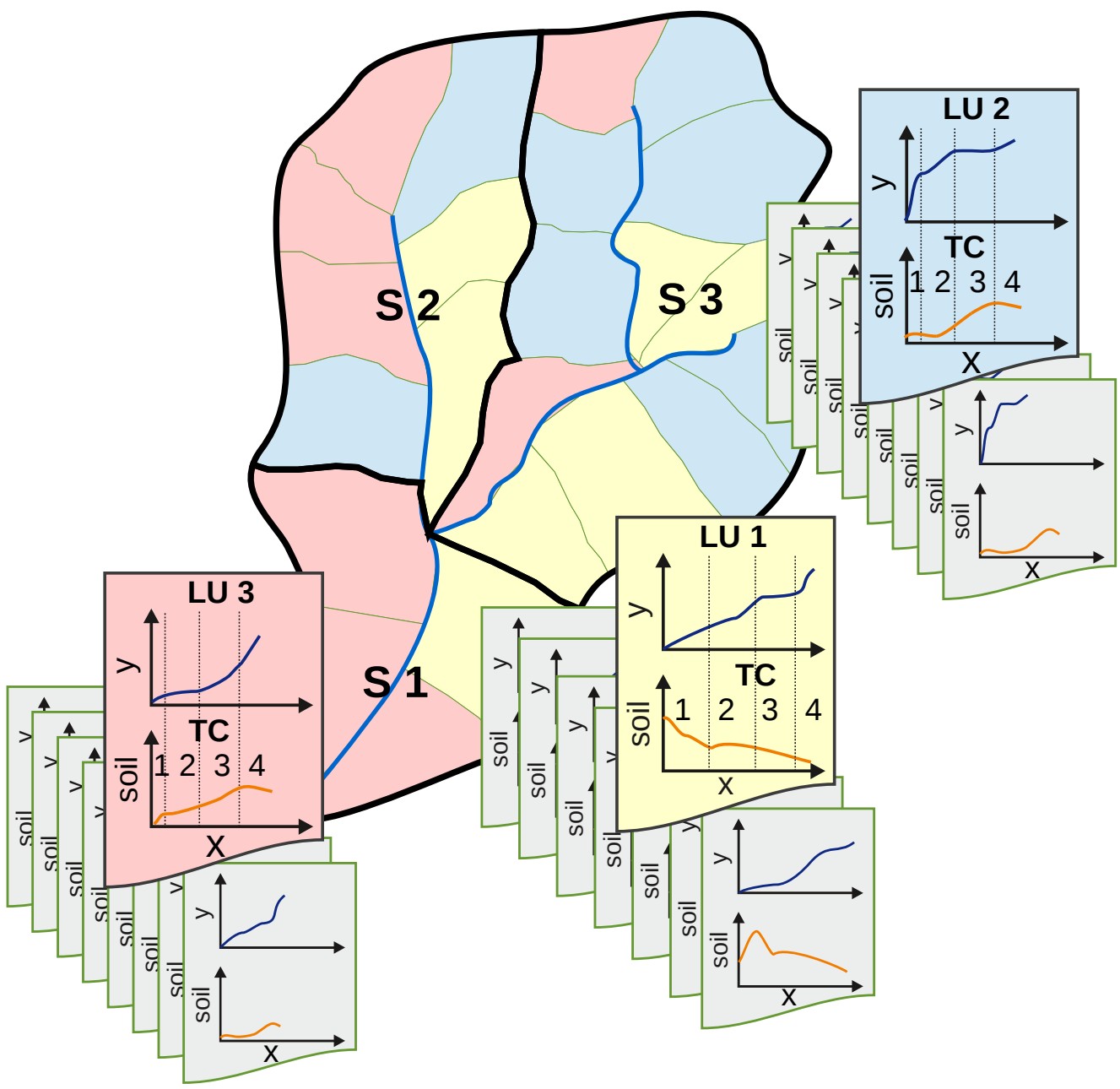

**Figure 2.** Example of the outcome of steps (i) to (iv) of landscape discretisation. Shown is a hydrological basin sub-divided into subbasins (labelled S 1–3), EHAs (small polygons), and LUs (red, blue, yellow) with TCs. Along-slope catena properties for the EHAs and LUs are summarised by the diagrams.

### 3.4 Parameter database

To store the results of landscape discretisation and allow easy maintenance, manipulation, and the generation of necessary model input files, all processing results are stored in a database. For flexibility, lumpR currently currently supports multiple DBMS, including MS Access, MySQL / MariaDB, and SQLite. Details for database configuration are provided in the Wiki of lumpR's web page (see Sect. 7). For interacting with the database, the package provides a set of functions. Again, these functions should be applied in the recommended order as indicated in Fig. 1 and will be further explained in the following.

*(i) db_create()* creates an empty parameter database. As the package underlies continuous further development, *(ii) db_update()* ensures backward compatibility to previous database versions. *(iii) db_fill()* assimilates the output of the landscape discreti­sation steps and other pre-processing not included in the package (e.g., readily prepared soil and vegetation parameters) and imports the data into the respective tables of the parameter database. *(iv) db_check()* performs a number of checks to identify and (if possible and desired) automatically resolve inconsistencies and/or missing information in the database. These include filtering of tiny and spurious areas (e.g., spatial entities smaller than a specified threshold) to reduce computational over­head, checking that all TCs have a slope larger than zero, defining special areas for separate treatment (this currently includes SVCs marked as water or impervious), removing special areas marked as water, computing fractions of impervious surfaces at TC level, removing impervious surfaces, estimating a storage coefficient for groundwater delay at LU level, deleting ob­solete datasets (i.e., unused spatial entities), checking for completeness (all IDs in the `*_contains_*` tables exist within the respective referenced tables), and computing the subbasin order from upstream to downstream. The user can decide, which checks to perform, how to deal with inconsistencies, and define thresholds for certain checks. Fore reproducibility, any changes to the database will be logged in table `meta_info`.

At any point during the processing the user can freely inspect and adjust the parameter database by means other than the functions provided by lumpR. In the current version, the package provides the function *db_wasa_input()* to convert the values of the parameter database into input files for WASA-SED. However, the user may as well export the values needed and compile the input files for any other model. Furthermore, upon request, export function for other models can easily be added.

### 4 Example application and sensitivity analysis

Within an example application, multiple realisations of hillslope discretisations of the same model domain are generated with lumpR version 2.0.0 by varying five parameters that control the creation of model entities on different scales. These realisations are then used with the hydrological model WASA-SED and their effect on the model output is analysed. In the following subsections, the study site, the WASA-SED model and data used for model initialisation, the sensitivity analysis of the discretisation parameters, and, finally, the results of this study are described.

## 4.1 Study site

To demonstrate the functionalities of lumpR the Benguê catchment was selected (Fig. 3). It is part of the upper Jaguaribe river catchment in the northeast of Brazil within the federal state of Ceará. The area has been investigated in a number of studies, in many cases employing the WASA-SED model, and was thus selected to ensure the suitability of the model for the catchment (Medeiros et al., 2010; Krol et al., 2011; de Araújo and Medeiros, 2013; Bronstert et al., 2014; Medeiros et al., 2014; Medeiros and de Araújo, 2014; de Figueiredo et al., 2016).

The Benguê catchment drains an area of about 926 $km^2$. At its outlet, the ephemeral Umbuzeiro river disembogues a reservoir built in 2000 with a storage capacity of 19.6 million $m^3$ to enhance water supply and reliability in the region. As the area is located within the "drought polygon" of Brazil, annual average precipitation is low with about 600 mm in comparison to a potential evapotranspiration of more than 2000 mm. The mean annual temperature is 25 °C with little variation. Climate is further characterised by a strong intra-annual variation of precipitation leading to distinct rainy (January to May with more than 80 % of annual rainfall) and dry seasons. Rainfall is mostly convective and concentrated in only a few high-intensity events per year.. Inter-annual variation of precipitation, however, is also high causing recurrent droughts which in cases may last over several consecutive years. The dominant natural vegetation is Caatinga consisting of deciduous bushland with xerophytic species ranging from dense dry forests to almost desert-like sites. The environment is further characterised by mainly sedimentary plateaus in the southern and western parts of the study site with steep terrain and deep (> 1 m) permeable soils, predominantly Latosols. In the North and East crystalline bedrock is prevailing with shallow Luvisols (< 1 m) causing high runoff coefficients. In alluvial zones, Planosols are dominant. The population density is low (6.4 inhabitants per $km^2$) with rural lifestyle. Parts of the area are used for small-scale farming including cattle-breeding and growing of maize and beans in particular.

## 4.2 The WASA-SED model

The WASA-SED model, revision 247 from 29 September 2016, is used to study the effects of different landscape discretisations realised with lumpR on simulated streamflow dynamics. WASA-SED is a deterministic, process-based, semi-distributed, time-continuous hydrological model. The model was first introduced by and is described in detail within Güntner (2002), with special focus on its application in semi-arid environments. It has been frequently employed in semi-arid areas such as northeastern Brazil (including the Benguê catchment; for references see Sect. 4.1), India (Jackisch et al., 2014) and Spain (Mueller et al., 2009, 2010; Bronstert et al., 2014).

The model incorporates the Shuttleworth-Wallace approach for evapotranspiration calculation over sparsely vegetated surfaces and an infiltration approach based on Green-Ampt accounting for Horton-type infiltration. Via the complex hierarchical spatial disaggregation scheme (see Sect. 2.2), lateral re-distribution processes as well as re-infiltration along a hillslope are considered while the model can still be applied over large scales (up to the order of magnitude of 100,000 $km^2$). Large strategic reservoirs can be represented in an explicit manner while smaller ones are treated as lumped water bodies of different size classes to efficiently account for water retention of many small reservoirs in a study region. The model has been subsequently

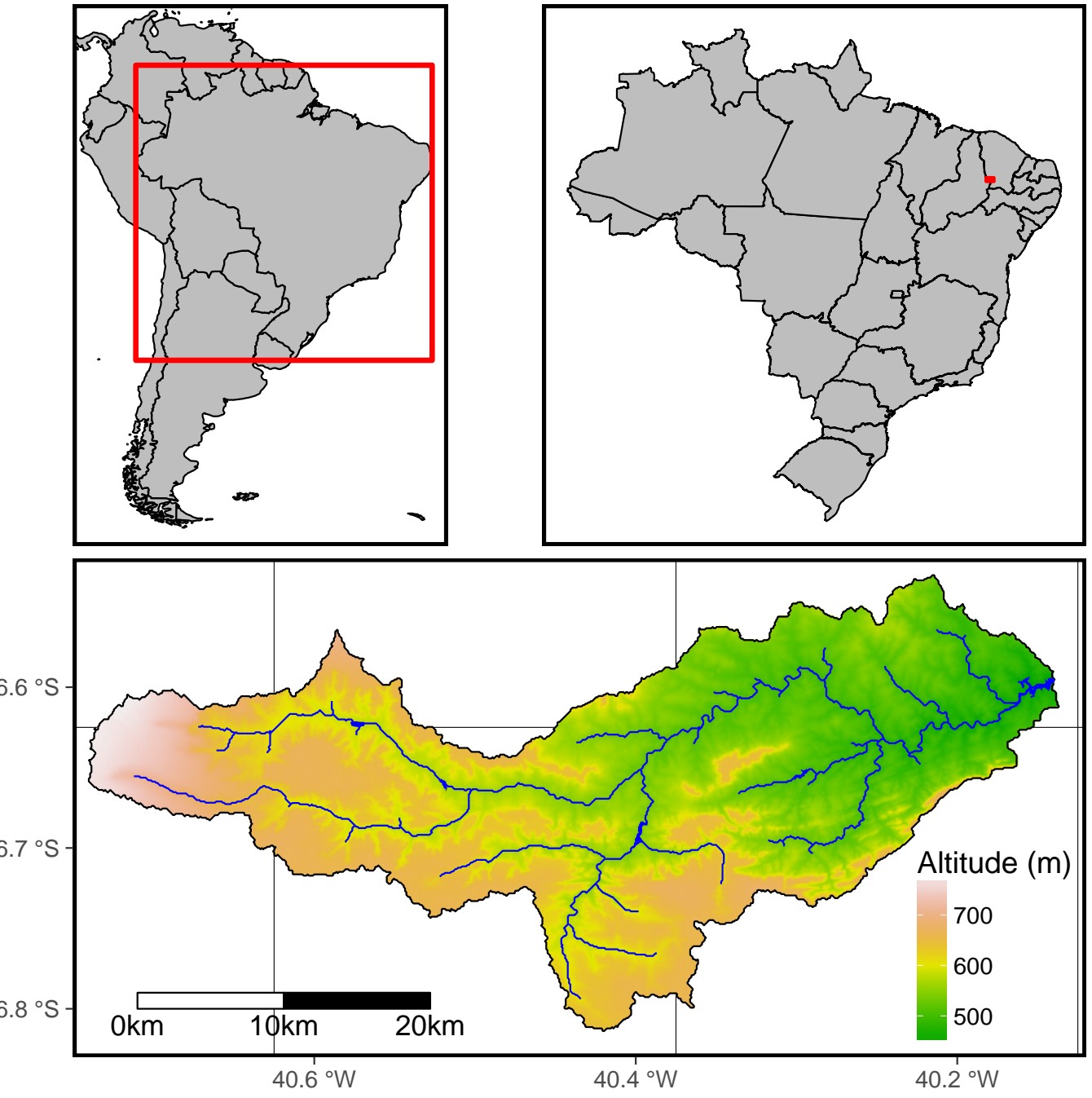

**Figure 3.** Overview over the Benguê catchment (lower panel) and location within Brazil (upper right) and South America (upper left), respectively. Black lines in the background of the lower panel correspond to the grid of the meteorological dataset of Xavier et al. (2016) that was used in this analysis (see Sect. 4.3).

expanded, e.g., to account for sediment dynamics and was renamed from WASA to WASA-SED (Mueller et al., 2010). It should, however, be noted that in this study erosion is not modelled.

## 4.3 Data and model setup

Meteorological data in daily resolution to drive WASA-SED, in particular precipitation, air temperature (obtained by minimum and maximum temperatures by simple averaging), relative humidity, and incoming short-wave radiation, have been derived from the dataset described by Xavier et al. (2016). This gridded dataset is based on station information that have been checked, corrected, and interpolated to a grid with 0.25° x 0.25° resolution. For the analysis, data from 20 grid cells were considered, whereas the catchment itself directly intersects with six cells (see Fig. 3). Extraterrestrial radiation as a further driver that has been calculated from astronomical relationships employing the R package *sirad*. Meteorological data have been interpolated to the locations of subbasin centroids using an inverse-distance weighting approach by employing the R package *geostat*.[1]

The basis of terrain analyses was the 90m x 90m SRTM DEM. As pre-processing step, the raw raster was sink-filled employing the GRASS function *r.terraflow*. Soil information have been derived from the database of Jacomine et al. (1973). The parameters needed by WASA-SED have been inferred by pedo-transfer functions based on soil texture information using the R packages *soilwaterfun* and *soilwaterptf* available via http://soilwater.r-forge.r-project.org/. Vegetation parameters for the types occurring within the study area have been elaborated during the development of WASA (Güntner, 2002) and have been adopted for this study. An updated shapefile of landcover distribution was obtained from the Brazilian Ministry for the Environment and the land cover classes have been reclassified to those used in (Güntner, 2002). Data on reservoirs and the geology of the area had been collected and processed within the SESAM project, see http://www.uni-potsdam.de/sesam.

No calibration of any model parameters has been done as a comparison with observational data shall not be part of this study. As has been mentioned, however, the model has been used in and proved its ability for the catchment (see references given in Sect. 4.1).

## 4.4 Sensitivity analysis of landscape discretisation parameters

During the process of landscape discretisation, a user commonly has to make a number of – often subjective – decisions. These are directly influencing the complexity of the discretisation and thereby affecting computational efforts and, possibly, model results. Although this issue has been acknowledged (e.g., Ajami et al., 2016; Fenicia et al., 2016), we are not aware of any study systematically analysing this effect for hillslope-based approaches and at multiple spatial scales at the same time. This may be mainly because of the associated manual effort and computational burden, which has become accessible using lumpR. Its fully automatic integration allows conducting a comprehensive numerical experiment, reflecting the complexity and multi-dimensionality in the discretisation process. Henceforward, we consider these as parameters within a model sensitivity analysis.

---

[1]This package is not on CRAN but available via the ECHSE tools library from https://github.com/echse/echse_tools.

**Table 2.** User decisions affecting landscape discretisation complexity and their realisations used for sensitivity analysis.

| Identifier | Meaning | Realisations |
|---|---|---|
| SUB_thresh | Minimum size of subbasins in number of grid cells | 1000, 2000, 5000, 10000, 30000 |
| EHA_thresh | Minimum size of EHAs in number of grid cells | 25, 50, 100, 200, 500, 750, 1000 |
| LU_no | Maximum number of LUs to be classified | 5, 10, 20, 50, 75, 100, 150, 200, 250, 300 |
| LU_atts | Number of attributes to be considered during LU classification | 1...7 |
| TC_no | Number of TCs to be deviated for every LU | 1...5 |

#### 4.4.1 Experimental setup

For landscape discretisation using lumpR five parameters reflecting the most important user decisions have been identified and are summarised in Tab. 2. Their presented realisations are based on expert knowledge. They result from a reasonable range of values while striving for maximum possible variation in the generated spatial units. Therefore, for some parameters such as `SUB_thresh`, also non-uniform distributions of the values have been taken into account. What follows is a reasoning on selected parameter realisations for the experiments that can as well be used as guidelines for lumpR applications.

`SUB_thresh` and `EHA_thresh` are size thresholds affecting the size and thereby the number of delineated subbasins and EHAs, respectively. As their realisations are given in number of grid cells, their choice depends on the resolution of the GRASS location which should be oriented on the DEM (here, the SRTM DEM resolution of 90m x 90m), and catchment size. `LU_no` and `LU_atts` control the process of clustering EHAs into LUs. The maximum possible value for `LU_atts` depends on the number of attributes that can be used for classification. These, by defaults, include the shape of EHAs, their horizontal and vertical extension, and a proxy for hillslope width which are all inferred from a DEM. Further supplemental attributes can be added which in this study included maps of soil types, land cover, geology, and SVCs that resulted in a total of seven attributes. In this study, different realisations of `LU_atts` thus simulate a differing degree of information available for the deviation of LUs. For `LU_atts` less than seven, the aforementioned attributes were sampled randomly. `LU_no` defines the maximum number of LUs to be generated. As the LU-classification is done successively for each attribute, this number results from the product of the number of classes $N_i$ specified for each of the `LU_atts` considered attributes $i$:

$$LU\_no = \prod_{i}^{LU\_atts} N_i \tag{1}$$

Thus, conversely, when `LU_no` is pre-specified, the values of $N_i$ need to be determined under the above-mentioned constraint as follows: One of the considered attributes is randomly selected and its $N_i$ increased by one. This procedure is repeated until Eq. 1 is satisfied, i.e., the actual number of LUs, is greater than or equal to `LU_no`. Finally, `TC_no` is the number of TCs that will be delineated for every LU.

**Table 3.** Streamflow indices used as scalar response functions for sensitivity analysis.

| Symbol | Index | Calculation | Unit |
|---|---|---|---|
| $RR$ | Runoff ratio | Sum of daily streamflow values divided by sum of daily precipitation over the whole period of analysis multiplied by 100 | % |
| $P_{flow}$ | Probability for significant streamflow | Number of days with significant[a] streamflow divided by total number of values multiplied by 100 | % |
| $Q_{avmax}$ | Average annual maximum flow | Average over all annual maximum streamflow values | $\mathrm{m^3 s^{-1}}$ |
| $SFDC$ | Slope of flow duration curve | Average slope of the flow duration curve for significant[a] medium ranged[b] streamflow values; high values stand for a more variable whereas low values represent a more damped flow regime (Sawicz et al., 2011) | dimensionless |
| $f_{low}$ | Frequency of low flows | Average number of insignificant[a] flow events[c] per year | $\mathrm{year^{-1}}$ |
| $f_{high}$ | Frequency of high flows | Average number of high flow[d] events[c] per year | $\mathrm{year^{-1}}$ |
| $RC_{rise}$ | Rate of change during rise | Average rate of change of the rising limbs of high flow[d] events[c] | $\mathrm{m^3 s^{-1} day^{-1}}$ |
| $RC_{fall}$ | Rate of change during fall | Average rate of change of the falling limbs of high flow[d] events[c] | $\mathrm{m^3 s^{-1} day^{-1}}$ |

(a) (In-) significant streamflow defined as those values (less than or equal to) larger than $0.01\ \mathrm{m^3 s^{-1}}$.

(b) Values between the 33 % and 66 % percentiles.

(c) An event is defined as a period of consecutive days a certain condition is fulfilled.

(d) High flows are those values being larger than a flow threshold which is herein defined as the 90 % percentile of all significant[1] flow values from all 12,250 model realisations during the analysis period.

For the sensitivity analysis all possible combinations of parameter realisations were employed which resulted in a total of 12,250 realisations of discretisations. These comprise varying complexities within all spatial levels and different degrees of data availability. Finally, WASA-SED was run with each realisation over a 13 year period, where the first five years were considered as warm-up and thus have been excluded from the analysis.

### 4.4.2 Scalar model output

The target variable of the analysis is the time series of simulated daily river contributions to the Benguê reservoir located at the catchment outlet. However, for conducting the desired sensitivity analysis, a scalar target function is needed. As it is impossible to summarise all important characteristics of a streamflow time series in a single scalar value, we employed multiple indices and performed the sensitivity analysis for each index separately. The indices are presented and described in Tab. 3. The indices were chosen to describe a wide range of aspects of streamflow behaviour ranging from the magnitude of flow ($RR$, $P_{flow}$, $Q_{avmax}$) over flow regime ($SFDC$) to frequency ($f_{low}$, $f_{high}$) and runoff concentration time ($RC_{rise}$, $RC_{fall}$).

### 4.4.3 Analysis method

Numerous approaches for sensitivity analysis exist (Pianosi et al., 2016). Their choice depends on the objective of the study, the nature and complexity of the model, its parameters and outputs, and available computing resources.

The goals of this analysis were, first, a *ranking* of the described descritisation parameters in terms of their influence (sometimes also referred to as *priorisation*) and, second, the identification of those parameters with negligible influence on the respective streamflow index (also referred to as *screening* or *fixing*). The above-mentioned sampling procedure for the parameters corresponds to a *global* sensitivity analysis with *all-at-a-time* sampling. This allows a variance- or density-based approach Pianosi et al. (2016). The former is based on the calculation of sensitivity indices based on the variance of the response function. This, however, requires the assumption that the variance is a good proxy for describing the variation of the value range. For multi-modal or highly-skewed distributions this cannot be guaranteed. In such a case Pianosi et al. (2016) recommend density-based methods. Rather than the variance alone this family of sensitivity analyses considers the probability density function of the response surface.

For the above-mentioned reasons, we chose the recently introduced PAWN method by Pianosi and Wagener (2015) as this density-based method can cope with skewed distributions and is relatively easy and straightforward to implement. PAWN uses empirical approximations of the unconditional cumulative distribution function $F_{y_i}(y_i)$, with $y_i$ being one of the eight streamflow indices selected as scalar response functions over all 12,250 realisations, and the conditional cumulative distribution functions $F_{y_i|p_j}(y_i)$ where a certain parameter $p_j$ is fixed at a specific value. The PAWN index $T_j$ as a sensitivity measure can then be calculated for each parameter employing a numerical approximation of the Kolmogorov-Smirnov statistic $KS$ following

$$KS(p_j) = \max_{y_i}\left|F_{y_i}(y_i) - F_{y_i|p_j}(y_i)\right| \tag{2}$$

and

$$T_j = \underset{p_j}{\mathrm{median}}\left[KS(p_j)\right] \tag{3}$$

$T_j$ varies between 0 and 1, where low values of $T_j$ identify the less influential parameters. For parameter screening the two-sample Kolmogorov-Smirnov test was employed. It calculates a critical value $KS_{crit}$ above which a parameter is significant as its conditional cumulative distribution function differs significantly from the unconditional one at a certain confidence level $\alpha$:

$$KS_{crit} = c(\alpha)\sqrt{\frac{n+m}{nm}} \tag{4}$$

with $n$ and $m$ being the number of samples to estimate $F_{y_i}(y_i)$ and $F_{y_i|p_j}(y_i)$, respectively, and taking the tabulated value of $c(\alpha) = 1.36$ for an $\alpha = 0.05$.

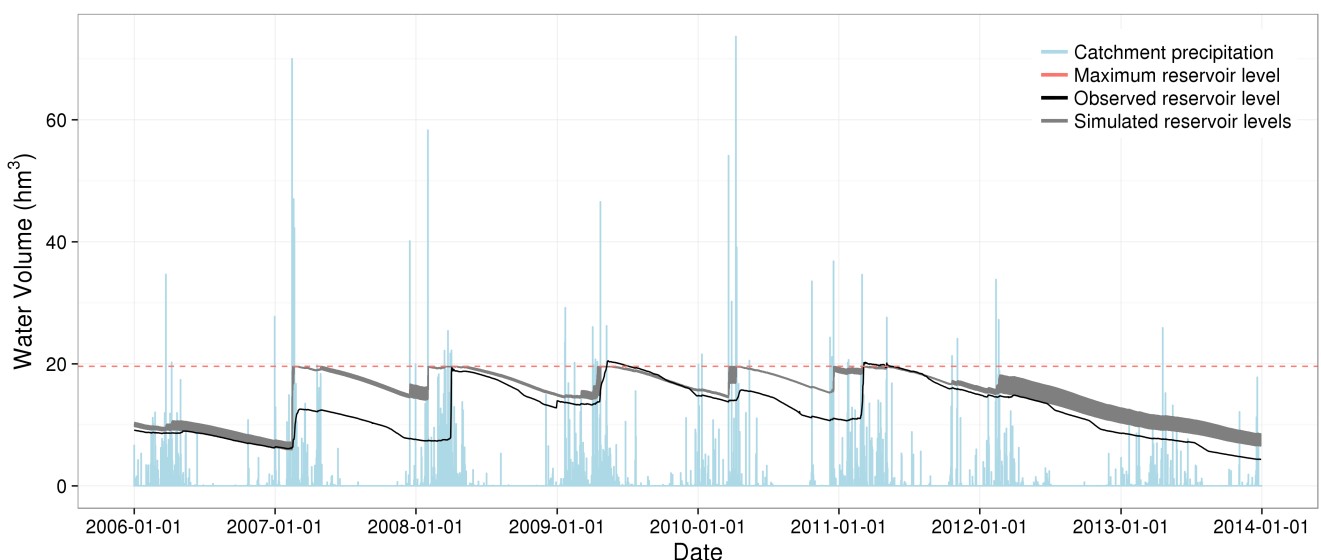

**Figure 4.** Time series plot of daily resolution of simulated (all 12,250 parameter realisations) and observed Benguê reservoir volume and areal precipitation. Note that the maximum reservoir level corresponds to the official value which has been used for model parameterisation. Observations sometime show slight deviations due to, e.g., random measurement errors or changed rating curves.

## 4.5 Results

Instead of the simulated river discharge (i.e., the model output used to calculate the streamflow indices, see Sect. 4.4.2), Fig. 4 provides an overview over the simulated reservoir storages in comparison to observations for the Benguê reservoir at the catchment outlet. We chose the latter because of the very episodic characteristic of the river discharge while the volumes, for visual comparison, constitute a more informative representation and are more directly related to available measurements. Figure 4 furthermore shows the catchment's areal precipitation used to drive the model. The absolute deviations of these uncalibrated model runs from the observations are, in parts, considerable, whereas qualitative behaviour is matched well. However, in some years (2008 and 2011) the model simulates reservoir filling within the rainy season much earlier than observed. Simulated reservoir depletion is often slower than it can be observed which might be the result from an imperfect parameterisation of reservoir abstractions for water consumption. Overall it should be noted that variability caused by different parameter realisations is rather small. Deviations mainly appear in a way that different discretisations result in slightly different amounts of generated runoff and, as a consequence, different reservoir level changes. Naturally, this cannot be observed during runoff events causing the maximum reservoir level to be exceeded in which additional runoff is lost as reservoir overspill. Furthermore, it can be observed that a rainfall volume of at least about 35 $\text{hm}^3$ (which is almost equal to 35 mm) seems to be necessary to produce noticeable reservoir inflow.

Figure 5 gives an overview of the streamflow index value distributions from the 12,250 realisations. Some indices, namely $Q_{avmax}$, $RC_{rise}$, and $RC_{fall}$, show distinct multi-modal value distributions. Distributions for the other indices are slightly

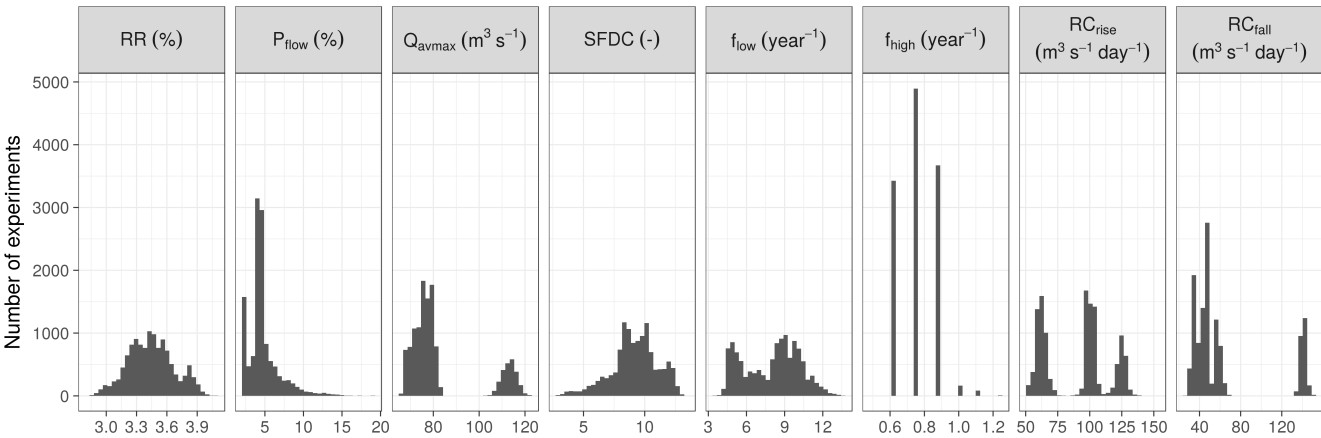

**Figure 5.** Histograms showing the value distributions of streamflow indices over the 12,250 realisations.

bi-modal ($f_{low}$) or skewed ($P_{flow}$ and $SFDC$) with $RR$ as the only index being more or less normally distributed. In general, the runoff coefficient $RR$ is consistently low ranging between 3 % and 4 % and streamflow can be characterised as ephemeral due to a low probability of days showing significant streamflow ($P_{flow}$ about 5 % for most of the experiments). Opposed to low flow periods, high flow events do rarely occur. The variability of $f_{low}$ is relatively high whereas $f_{high}$ shows relatively
low variance. Peak flows ($Q_{avmax}$) as well as runoff concentrations (characterised by $RC_{rise}$ and $RC_{fall}$) vary considerably between the realisations also exhibiting multi-modal distributions.

Screening and ranking of landscape discretisation parameters is illustrated in Fig. 6. The size of subbasins (`SUB_thresh`) is the most influential parameter for all indices except for those being related to low flow characterisation ($f_{low}$ and $P_{flow}$) which are dominated by the size of EHAs (`EHA_thresh`). The number of LUs (`LU_no`) can be regarded as the third important
parameter being especially of relevance for high flow related indices ($f_{high}$ and $Q_{avmax}$) and, to some extent, flow regime ($SFDC$) and runoff concentration ($RC_{fall}$). The least important parameters are the number of TCs (`TC_no`), except for runoff concentration ($RC_{rise}$ and $RC_{fall}$), and the number of attributes considered for LU classification (`LU_atts`) which is insignificant for all streamflow indices.

More information on parameter influences on the various streamflow indices can be obtained from Fig. 7. The more influen-
tial a parameter, the larger the deviations of the conditional empirical cumulative distribution functions from the unconditional one. Many of the diagrams show a clear relationship between parameter realisation and streamflow index value. The larger the subbasins (i.e., the larger `SUB_thresh` and the lower the number of subbasins) the smaller the generated amount of runoff ($RR$ gets smaller) and the smaller the probability of significant runoff ($P_{flow}$). On the other hand, peak discharges ($Q_{avmax}$) increase and the catchment appears to produce more rapid runoff responses (higher values of $RC_{fall}$ and $RC_{rise}$). Furthermore
it can be seen that `SUB_thresh` is responsible for the multi-modal distributions of $Q_{avmax}$, $f_{high}$, $RC_{fall}$, and $RC_{rise}$ as the conditional distribution functions show a less stepped shape than the unconditional functions. The influence on $f_{low}$ appears to be less clear. It can be seen, however, that larger values of `SUB_thresh` result in a more pronounced bimodal distribution of

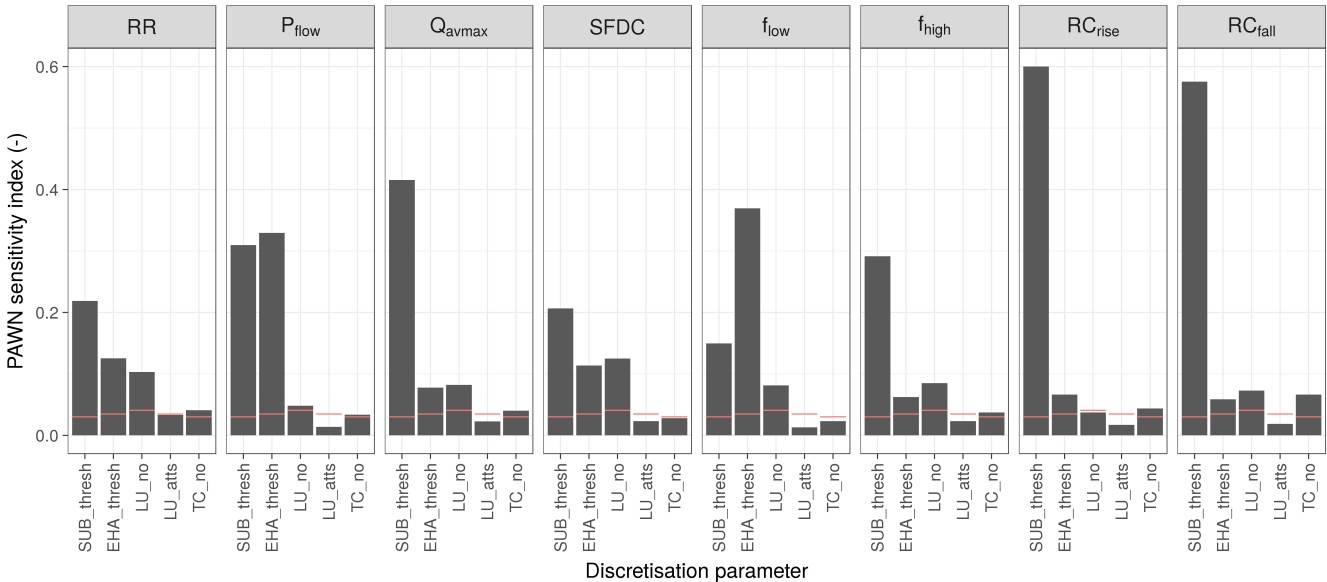

**Figure 6.** Barplots of PAWN sensitivity indices $T_j$ for each streamflow index and landscape discretisation parameter. Red lines indicate critical Kolmogorov-Smirnov values ($KS_{crit}$), i.e., parameters, for which $T_j$ is not greater than this value, can be regarded as insignificant for the respective streamflow index.

that index. Low flows are otherwise more dominated by the size of EHAs (`EHA_thresh`). The larger `EHA_thresh` (i.e., the larger the EHAs and the lower their number) the less the probability for significant streamflow ($P_{flow}$) and the less the number of low flow events per year ($f_{low}$) while the flow regime becomes more variable (higher values of $SFDC$). Higher numbers of LUs (`LU_no`) result in more generated runoff ($RR$ increases) with a tendency to both higher frequencies of low flow and high
5  flow events, and generate more variable streamflow regimes.

## 5  Discussion

### 5.1  lumpR: features, benefits, limitations

It is still common practise for many researchers in the field of hydrological modelling to not automate their pre-processing steps. Even if they do, the related scripts are rarely published along with the studies. Furthermore, common limitations of
10  existing software are that they are often model specific, and/or perform only certain steps of pre-processing. Some tools are commercial or can only be used along with commercial software (e.g., ArcGIS or MATLAB).

With lumpR, a package for the free and open-source programming language R has been developed that addresses these limitations and build on the philosophy of FOSS. So far, the software has been tested under Windows as well Linux-based operating systems (openSUSE and Ubuntu). lumpR interacts with the GIS GRASS and thus allows graphical investigation

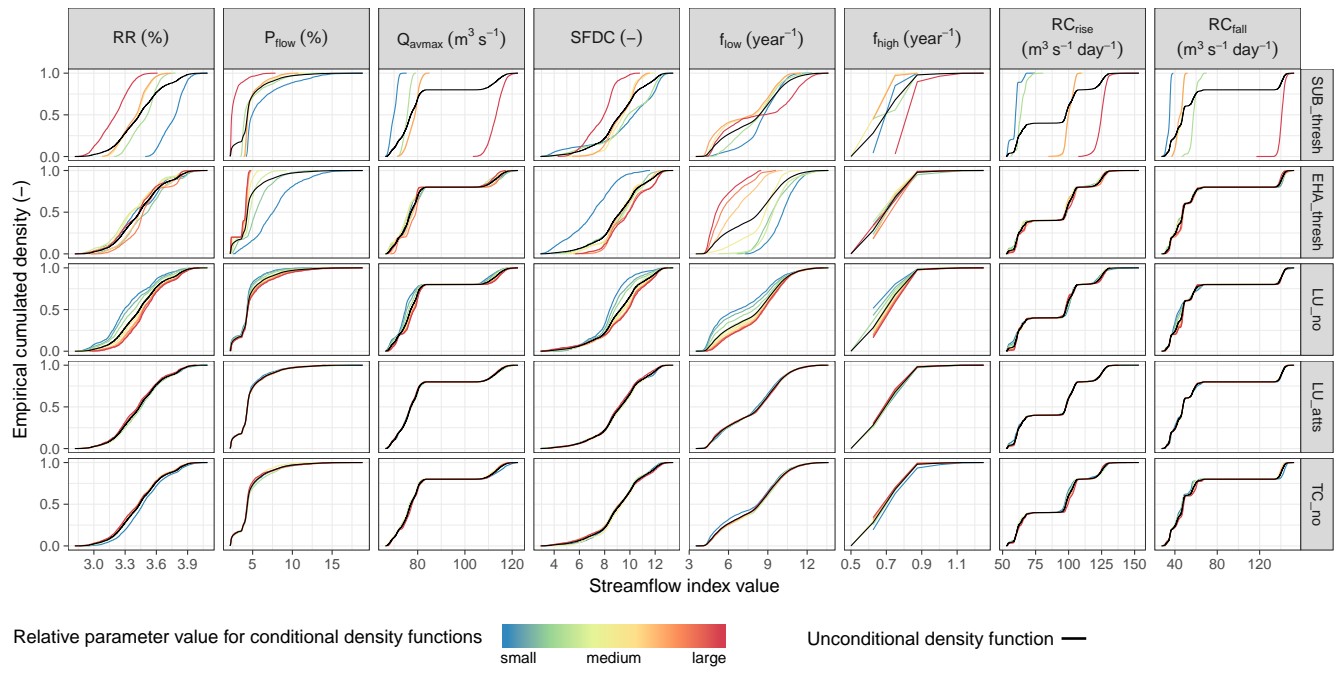

**Figure 7.** Unconditional (black lines) and conditional (rainbow coloured lines) empirical cumulative distribution functions for each stream-flow index–discretisation parameter combination.

and manual correction of outcomes. As R is a widespread scripting language, model pre-processing in that way can easily be customised, automated and reproduced. Via the database tools, the software allows to keep the output directories clearly arranged by putting all information into a database. Several database systems are supported.

In this study, lumpR's functionality was demonstrated together with the model WASA-SED. However, the package provides a
5  function to create the model's input files, which can by easily adapted to the requirements of other hillslope-based models. First tests employing the simulation environment ECHSE (Kneis, 2015) revealed the package to be easily adaptable for producing the input files needed by a different model. Other candidate models are, in principal, all hydrological models with similarly complex spatial aggregation schemes as used by WASA-SED that can make use of the information provided by lumpR, such as the WEPP model (Flanagan and Nearing, 1995).[2]

10  The centrepiece of the package is the LUMP algorithm introduced by Francke et al. (2008) for the calculation of hillslope properties and the delineation of representative LUs together with the subdivision into TCs. In contrast to simple GIS overlay techniques, as usually employed for the delineation of HRUs, it preserves information on the distribution of hillslope parameters and their relative topographic position, i.e., their downslope connection. In that way, the task of hillslope-based landscape discretisation and parameterisation can be purposefully directed to landscape properties dominating hydrological response.
15  Via its integration into lumpR, the application of the algorithm could be further simplified and harmonised as the original

---

[2]Readers considering lumpR for use with their model are encouraged to contact the corresponding author of this paper for support.

LUMP consisted of a loose collection of less user-friendly scripts, partly also relying on non-free software such as MATLAB. In that way, the package hides unnecessary detail from the user while at the same time ensuring a certain level of control over the discretisation process. For instance, in the present study, the use of morphological parameters was limited to shape, horizontal and vertical extension, and hillslope width. The lumpR package is, however, flexible enough that a user can include further parameters as supplemental information for the classification process. This might include the use of other factors relevant for hillslopes characterisation such as, for instance, contour curvature being related to the convergence or divergence of flowpaths and as such being of hydrological relevance (Bogaart and Troch, 2006). Furthermore, lumpR comes with a lot more additional functionalities than the mere LU and TC deviation.

Thanks to lumpR allowing a high degree of automation, for the first time a multi-hierarchy sensitivity analysis of discretisation parameters in a hillslope model could be conducted. Within this analysis, a high number of discretisations of varying complexity were easily produced. Thus, a user can experiment and find out the optimal degree of complexity for a certain catchment and a specific objective, e.g., by systematically employing a multiple hypothesis framework. For the presented case study, the analysis revealed the pronounced influence of the size of the subbasins and the EHAs on various aspects of the hydrograph. All other discretisation parameters showed no or considerably less influence.

During the testing phase, some shortcomings of lumpR were identified. With regards to the applicability over large datasets, i.e., when applying the package to large areas in the order of $>100,000$ $km^2$ and/or when employing high-resolution DEMs, time consumption might pose a restriction, although lumpR already uses parallelised code in the most critical steps. Therefore, future enhancements also need to include further improvements regarding computational efficiency.

A limitation more related to the algorithm is that the software is not able to automatically distinguish and account for artificial hydrological discontinuities. This includes, e.g., ditches and field boundaries or other problematic formations such as large flat areas in a DEM as produced by lakes. While the former pose restrictions on the general applicability of the hillslope approach, the latter need to be masked in GRASS before the analysis. In addition, some of the pre- and post-processing steps within lumpR (i.e., functions *lump_grass_prep()* and *lump_grass_post()*) still employ rather simplistic approaches. This affects in particular the deviation of the river network, subbasin delineation, and the approximation of streamflow routing parameters (the latter tailored to the rather simplistic unit hydrograph approach of WASA-SED). In this respect, future enhancements should also include a review on latest advancements of terrain analysis and parameterisation and the refinement of employed algorithms.

## 5.2 On the sensitivity analysis of discretisation parameters

As in science reproducibility and objectivity are primary criteria for any investigation, it has to be noted that any model discretization is subject to a certain degree of subjectivity. Especially for hillslope-based discretization, this can cover several hierarchy levels. Consequently, the effects of these choices on the model output have been assessed via the sensitivity analysis within the example application and its results shall be discussed in the following.

The results from the 12,250 realisations of landscape discretisation show only little difference with respect to water storage dynamics of the Banguê reservoir (Fig. 4). The small variation of the runoff coefficient (see Fig. 5) further supports the

conclusion that decisions on landscape discretisation parameterisation only have a minor impact on simulated runoff volume for the given case. On the other hand, the influence on other indices describing runoff concentrations and dynamics, and the frequency of flood or drought events is much more obvious.

The hydrological regime of the study area is primarily influenced by precipitation, which is characterised by a high temporal concentration and a large temporal and spatial variability. A comparison with the data reported in Medeiros and de Araújo (2014) further supports that uncertainties regarding the precipitation input to the model have a much larger impact on simulation results than the discretisation parameters. In their study, Medeiros and de Araújo (2014) used a set of raw station data in contrast to the preprocessed and gridded dataset by Xavier et al. (2016) used for our experiments and their runoff values have been assessed by taking the Benguê reservoir inflows computed from water balance calculations. Their reported runoff coefficients are mostly lower than ours, even when precipitation is higher, and shows less inter-annual variation (see Fig. A1 in Appendix).

The precipitation forcing for the current study was implicitly slightly influenced by variable subbasin sizes and numbers, as the precipitation was specified at the subbasin level. The variability in precipitation input among the realisations, however, appears to be negligible as it is generally less than 6 mm for daily values and less than 10 mm for yearly sums (see Fig. A2 in Appendix).

Our simulation results show a general overestimation in comparison to measured values (Fig. 4). Despite the mentioned uncertainties arising from the precipitation input, there are some other possible factors that could have led to the observed mismatch: uncertainties in the reservoir parameterisation in the model (e.g., we use a static parameterisation of reservoir abstractions which are, in reality, dynamic); uncertainties in the observations (e.g., due to deficiencies or changes of the rating curve); model parameterisation uncertainty (the model has been run with standard parameterisation for the area without further calibration, see Sect. 4.3). Regarding the parameterisation it should furthermore be noted that the different discretisations did not directly affect soil nor land-cover parameters. They merely modified the fractions of soil and vegetation types that are assigned to the spatial units.

Considering the method of sensitivity analysis it can be concluded that the choice for a density-based approach was reasonable as most of the analysed streamflow indices exhibit multi-modal or skewed value distributions. A drawback of the analysis approach is that only first order effects of parameter sensitivities have been quantified while interactions among parameters have been neglected. It might thus be that insignificant parameters (i.e., the parameter `LU_atts`) have significant higher order effects due to parameter correlations. With respect to the discretisation parameterisations, it can be argued that the chosen parameter realisations are both subjective and case study specific. On the other hand, all parameter realisations show a monotonous effect on the streamflow indices, i.e., when increasing a sensitive parameter the streamflow index values increase or decrease monotonously (see Fig. 7). This suggest a continuous response of the parameters, facilitating some transferability of the results.

Overall, the example study and sensitivity analysis is catchment and model specific. Strictly speaking, conclusions are thus limited to applications of the same model under similar hydro-climatic conditions, i.e., semi-arid areas without substantial groundwater influences mainly characterised by spatially and temporally heterogeneous precipitation patterns. It remains an open question whether the use of a different model and/or the application in a catchment with distinct environmental and

climatological characteristics and/or different dominant runoff generation mechanisms would lead to other conclusions. This paper presents a novel framework along with an example application to address these questions in future studies.

## 6    Conclusions

The goal of this study was to introduce a new software for landscape discretisation in semi-distributed hydrological modelling. Thereby, three objectives have been pursued.

Firstly, we provided a short review of existing landscape discretisation algorithms and software solutions. The number of existing concepts and corresponding tools was found to be large, making it a difficult task to choose a specific approach and software. Besides grid-based approaches, the most common strategies for semi-distributed hydrological modelling focus on the delineation of spatial entities with homogeneous process dynamics, such as the frequently implemented HRU approach. Approaches directly concentrating on the description of a hillslope as central modelling unit or pursuing hierarchical multi-scale frameworks as efficient solutions for large-scale application are less in number. In addition, existing programmes implementing a specific discretisation often exhibit various limitations, e.g., they are model specific, commercial or employ commercial back-end software, or allow only a limited or no automation of workflows.

Secondly, we developed and presented a new software called lumpR as a package for the open source environment R. It was designed to implement a hillslope-based hierarchical multi-scale discretisation of landscapes, including the delineation of subbasins, the derivation and lumping of hillslopes, and the subdivision of the latter into terrain and soil-vegetation components. The package thereby connects to GRASS GIS, directly using prepared spatial information and writing spatial output into an initialised location for immediate inspection. Furthermore, database functionalities have been included to manage the outcomes of the discretisation process. lumpR overcomes existing limitations in a way that it easily allows to include different hillslope-based models, it is completely free and open source, and it facilitates the automation of workflows. At the same time, however, it is retaining a sufficient degree of freedom to the user via the selection of parameters, and the inclusion of expert knowledge and additional information.

Thirdly, the functionality of the package was shown in a case study in the semi-arid northeast of Brazil employing the hydrological model WASA-SED. Thereby, the workflow automation allowed a systematic sensitivity analysis of crucial landscape discretisation parameters. Regarding multiple streamflow metrics, the model appeared to be reasonably robust to the dicretisation parameters. The size of subbasins and delineated hillslopes were found to be the most influential factors. The number of landscape units (i.e., lumped hillslopes) and the further subdivision into terrain components appeared to be less important, the amount of information included in the hillslope lumping process being even completely insignificant.

The R package turned out to be an efficient and user-friendly tool for the automation of landscape discretisation for hillslope-based large-scale hydrological models. Future work, on the one hand, will focus on comparing uncertainties arising from discretisation to other sources of uncertainties. On the other hand, in order to obtain more general conclusions, the presented sensitivity analysis of landscape discretisation parameters needs to be extended to other catchments within different environmental and hydro-meteorological conditions as well as other hillslope-based models. Technical extensions will include the

integration of further models, improvement of time consumption and memory handling for application in large areas > 100,000 $\mathrm{km}^2$, consideration of artificial discontinuities and mechanisms for large flat areas, refinement of certain parameter estimation approaches, and testing the package for other hydro-meteorological and environmental conditions.

## 7   Code availability

Code for lumpR is freely available at https://github.com/tpilz/lumpR. The Latex code to reproduce this paper including R code to reproduce all analyses and figures is available at https://github.com/tpilz/lumpr_paper.

## 8   Data availability

Meteorological data are available from http://careyking.com/data-downloads/. DEM raw data can be obtained via http://srtm.csi.cgiar.org/SELECTION/inputCoord.asp by selecting tile 28/14 (horizontal/vertical). Reservoir data and the geology map have been processed within the SESAM project and are not publicly available. For more information and contact details see http://www.uni-potsdam.de/sesam. Land cover and soil raster maps are not publicly available.

## Appendix A: Precipitation uncertainty and comparisons with other studies

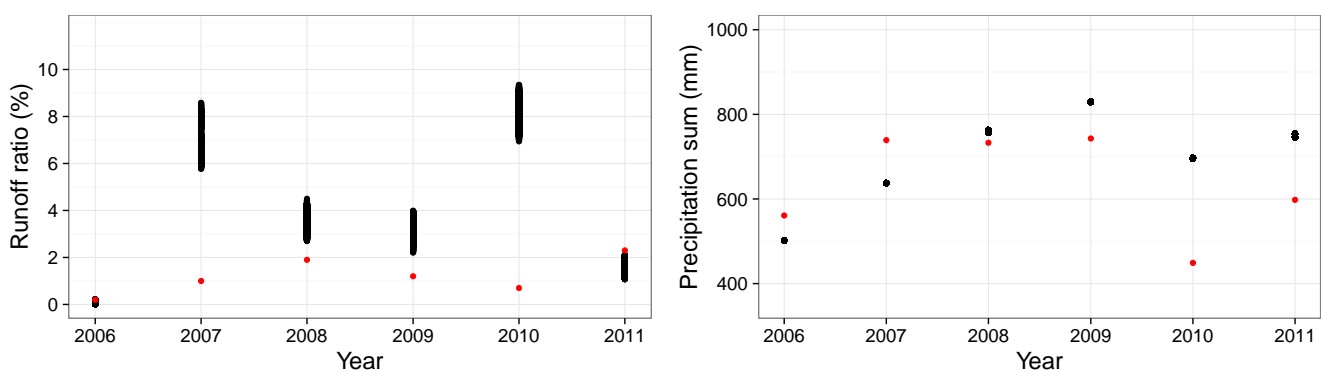

**Figure A1.** Comparison of simulated yearly runoff coefficients (left) and precipitation forcing (right) for all 12,250 discretisations (black dots) with values reported by Medeiros and de Araújo (2014) (red dots).

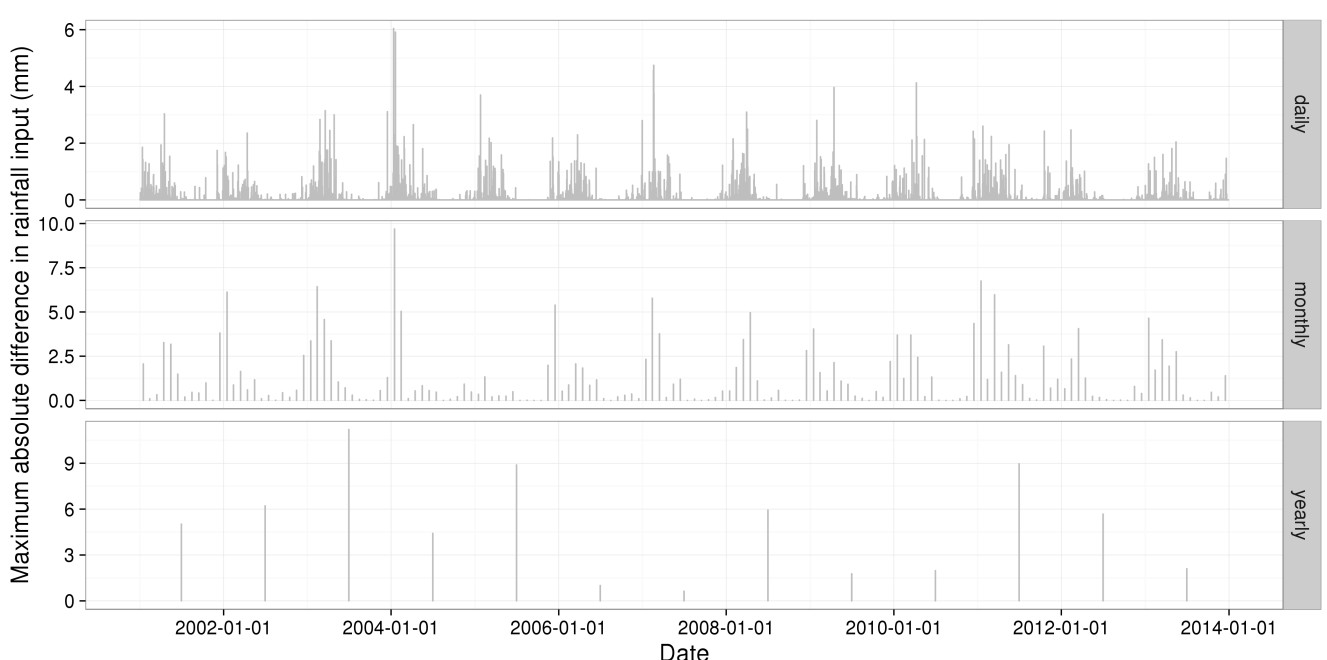

**Figure A2.** Maximum of absolute differences in rainfall input between the 12,250 discretisations for daily, monthly, and yearly aggregated time steps.

*Author contributions.* All authors contributed to the idea and methodology. T. Pilz and T. Francke developed the lumpR package (more specific contribution information are included in the source code files). Experiments and analyses have been conducted by T. Pilz with support by T. Francke. The manuscript was prepared by T. Pilz with contributions from all co-authors.

*Competing interests.* The authors declare that they have no conflict of interest.

5 *Acknowledgements.* Authors thank Fernanda Scholz (former M.Sc. student at the Leibniz University of Hannover) for the preparation of soil data. Student Lisa Berghäuser from the University of Potsdam is acknowledged for testing lumpR and giving valuable feedback and bug reports. Furthermore, we appreciate the feedback and suggestions on sensitivity analysis by Dr. Gabriele Baroni. Tobias Pilz acknowledges funding by the Helmholtz graduate school GeoSim.

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
