# Peer review of "lumpR 2.0.0: An R package facilitating landscape discretisation for hillslope-based hydrological models"

_Geoscientific Model Development, 2017_

## Referee Comment (RC1) · Anonymous Referee #1 · 12 Feb 2017

General comments: This manuscript introduced a computer software lumpR that serves as pre-processing tool for the hydrological model WASA-SED. This manuscript is more like a technical document, it needs to emphasize the key features and major functions of this software. It also declared that "the first objective of this paper is to provide an overview of existing landscape discretisation algorithms and software", however this review need a major revision.

Major comments:

There are many literature reviews in each sections. Please move these literature reviews into Section 1 or/and Section 2.

Section 1: Introduction

1) Classification of the hydrological modelling approaches into three types (fully distributed approach, lumped approach and semi-distributed approach) is often questioned. For example, many large scale hydrological model uses grid cells for the discretisation of the landscape, but the size of the grid cell is very large, so they also employ sub-grid parameterization schemes. What type do these hydrological models belong to?

2) The logic between the literature reviews in Section 1 and Section 2 is not clear.

3) The logic between the first objective and the second objective is not clear. Before the the overview of existing landscape discretisation algorithms, authors already determined to use hillslope-based approach.

Section 2: Review of landscape discretisation in hydrological modelling

1) This section should focus on the landscape representation, not only the landscape discretization.

2) Section 2.1 presents common knowledge about the DEM. What is relationship with section 2.2?

2) To model the catchment hydrology, water flow pathway is a key issue in landscape discretization/representation. This section lacks of representation of catchment water flow pathway. (Reference: Yang D, Gao B, Jiao Y, et al. A distributed scheme developed for eco-hydrological modeling in the upper Heihe River. Science China Earth Sciences, 2015, 58(1): 36-45.)

3) What are the basic units of hydrological simulation in different landscape discretization approaches (different semi-distributed hydrological models)?

4) In Section 2.3, it's hard to see what kinds of software is suitable for landscape discretization?

Section 3:

1) A flowchart is needed in description of lumpR. Figure 1 should be modified to contain both discretization procedure and major functions.

2) This section should be rearranged, for example, general workflow, major functions, additional tools.

3) How to define the topological relationships between the hillslopes and sub-catchment?

4) As shown in Figure2, how to simulate the flow discharge in the river networks? It's also important to consider the spatial variability of precipitation inputs.

5) This is the main part of this MS, more detailed information is required.

Section 4:

1) Sensitivity analysis in this section is repeated in Section 5. I suggest move the sensitivity analysis to Section 5.

2) This section is very confusion. It's very hard to see how to implement the lumpR for application of WASA-SED model. Please rearrange this section.

3) What are the inputs (climate forcing data) and outputs of WASA-SED model?

4) What are the major functions of WASA-SED model? Please show these functions by using the simulated results.

Other comments:

1) Yang et al. (2002) developed a hillslope-based hydrological model. This work should be cited. Yang, D, S Herath & K Musiake (2002), A Hillslope-based hydrological model using catchment area and width functions. Hydrological Sciences Journal, 47(1), 49-65.

2) Some references should be added in Section 1, e.g. Yang et al., (2000), Comparison

of different distributed hydrological models for characterization of catchment spatial variability. Hydrological Processes, 14, 403-416.

3) P4, L32-33: Here should emphasize that the models use HRUs face the difficulty of representing water flow pathways (for example the runoff from hillslopes and runoff along the river networks) appropriately.

4) P5, L15-20: The difficulty of determining the size of REW should be mentioned.

5) P6, L15: A table is needed to summarize the existing software for model pre-processing.

6) P12: A table is required to summarize the model parameters.

7) P19: The discharge hydrograph is required in addition to the reservoir water volume shown in Figure 4.

8) P 20: Use a table instead of Figure 5.

9) P 21: Show the results of sensitivity analysis using the hydrograph too.

---

## Short Comment (SC1) · 24 Feb 2017

Dear authors,

in my role as Executive editor of GMD, I would like to bring to your attention our Editorial version 1.1:

http://www.geosci-model-dev.net/8/3487/2015/gmd-8-3487-2015.html

This highlights some requirements of papers published in GMD, which is also available on the GMD website in the 'Manuscript Types' section:

http://www.geoscientific-model-development.net/submission/manuscript_types.html

In particular, please note that for your paper, the following requirement has not been met in the Discussions paper:

- "The main paper must give the model name and version number (or other unique identifier) in the title."

Please add a version number for lumpR in the title upon your revised submission to GMD.

Yours,

Astrid Kerkweg

---

## Referee Comment (RC2) · Anonymous Referee #2 · 3 Mar 2017

General comments: Authors have developed an R package, lumpR, for watershed discretization in hillslope-based hydrological models. While developing an open source package for catchment discretization is the main goal of this manuscript, this paper does not introduce any new algorithm for hillslope delineation and semi-distributed hydrologic modeling that is superior to the existing approaches. As discussed in Section 2.3, series of SHELL and MATLAB scripts are already available to process data for the LUMP modeling approach. As a result, most of the paper reads like a user manual for the R package.

On the other hand, the sensitivity experiment described as the case study in this paper is the strongest component of this paper. Therefore, I suggest authors to expand on

the sensitivity experiment by considering some of the limitations they discuss as well as examining the impact of catchment discretization on other components of the water budget.

How general are the findings from the sensitivity experiments? Do you expect to obtain similar results if you implement the approach in other basins?

Major comments:

1) Page 8-L10- Authors mention that they implemented the LUMP algorithm as the basis for development of the R package. However, it is not clear whether the semi-distributed approach that is implemented in the LUMP model is superior to other discretization approaches in other semi-distributed modeling approaches. What are the advantages of using the LUMP approach compared to other discritization approaches?

2) Please switch Figure 1 with Figure 2 as it makes sense to have the discretization approach first and then the software structure.

3) Page 10-L10-Could you please explain Environmental Hillslope Areas?

4) In Section 3.2-It will be great if you can refer to different components of Figure 2 as you explain various functions.

5) Page 10-L20-What criteria are used to further subdivide each LU to terrain components?

6) Figure 4- It seems reservoir volume is always overestimated compared to observations for various sensitivity runs. Can authors explain the reason for this behavior?

7) Figure 6- It is not clear how general is the results of this figure. Do authors predict that similar sensitivity pattern will be obtained if a different hydrologic model is used?

8) Additional information about soil and landcover parameters are required and how discretization approach impacted parameter selection for each hillslope.

---

## Author Comment (AC1) · 14 Mar 2017

Dear Mrs. Kerkweg,

indeed we forgot to add the version number of the software in the title of the paper. We apologise for this and we will add the version number in the revised version of the manuscript.

Yours,

Tobias Pilz (on behalf of all co-authors)
* * *

---

## Author Comment (AC2) · 14 Mar 2017

Dear Reviewer,

thank you for your time and effort spent on reviewing our manuscript. In the following we will answer your comments point-by-point.

General comments: This manuscript introduced a computer software lumpR that serves as pre-processing tool for the hydrological model WASA-SED. This manuscript is more like a technical document, it needs to emphasize the key features and major functions of this software. It also declared that "the first objective of this paper is to provide an overview of

existing landscape discretisation algorithms and software", however this re-
view need a major revision.

We acknowledge the suggestions regarding the structure and readability of the
manuscript. We will revise the paper in order to better emphasise the key features and
functionalities of the presented software. Furthermore, we will provide a major revision
of the literature review (section 2) along with a summary of the analysed literature in
the form of a graphic or a table.

> There are many literature reviews in each sections. Please move these
> literature reviews into Section 1 or/and Section 2.

We agree that literature reviews should be rather concentrated. That is why we con-
centrated our literature reviews in the manuscript in Sects. 1 and 2. The cited literature
in the other sections is not part of a review, instead to refer to the specific algorithms
implemented in the software (Sects. 3.2 and 3.3), introduce the study site and the em-
ployed hydrological model (Sects. 4.1 and 4.2, respectively), citation of data sources
(Sect. 4.3), to explain the method used for sensitivity analysis of the software's param-
eters (Sect. 4.4), and to discuss the obtained results along with other studies (Sect. 5).
We would therefore kindly ask for a more specific description of this concern.

> Section 1: Introduction
>
> 1) Classification of the hydrological modelling approaches into three types
> (fully distributed approach, lumped approach and semi-distributed ap-
> proach) is often questioned. For example, many large scale hydrological
> model uses grid cells for the discretisation of the landscape, but the size
> of the grid cell is very large, so they also employ sub-grid parameterization
> schemes. What type do these hydrological models belong to?

We agree that the classification approach of hydrological models into the mentioned three categories can be ambiguous and is not always well-defined. We will therefore adapt the introduction in a way pointing out possible ambiguities in the conception of the classification in fully distributed, lumped approach and semi-distributed discretisations. We will furthermore explicitly refer to sub-grid parameterisation as another strategy of increasing the degree of detail in models without increasing the spatial resolution.

> 2) The logic between the literature reviews in Section 1 and Section 2 is not clear

In the introduction (Sect. 1), a classification of landscape discretisation concepts in hydrological modelling into fully distributed, semi-distributed, and lumped approaches is made while briefly explaining advantages and limitations of each concept. In the review (Sect. 2), however, the focus solely lies on semi-distributed approaches. In the revised version of the manuscript, we will focus on better pointing out that objective.

> 3) The logic between the first objective and the second objective is not clear. Before the the overview of existing landscape discretisation algorithms, authors already determined to use hillslope-based approach.

The first objective of the paper is to present existing landscape discretisation algorithms and software solutions along with their limitations. We introduced, as second objective, a new software picking up the identified limitations. Our intended logic between the two was to demonstrate the adequacy of presenting yet a further software. However, we will try to clarify the point in the revised manuscript.

> Section 2: Review of landscape discretisation in hydrological modelling

> 1) This section should focus on the landscape representation, not only the landscape discretization.

Thank you, we will adapt the title of Sect. 2. Indeed, the review is already focussing on landscape representation as a whole rather than the mere discretisation procedure.

> 2) Section 2.1 presents common knowledge about the DEM. What is relationship with section 2.2?

We acknowledge that Sect. 2.1 currently stands a bit apart from Sect. 2.2. Our intention was to first give a short overview over different topography representation approaches before discussing landscape discretisation as the former are the source of data for any landscape analysis and discretisation. In the revision we will try to establish a more profound relation between the two subsections.

> 2) To model the catchment hydrology, water flow pathway is a key issue in landscape discretization/representation. This section lacks of representation of catchment water flow pathway. (Reference: Yang D, Gao B, Jiao Y, et al. A distributed scheme developed for eco-hydrological modeling in the upper Heihe River. Science China Earth Sciences, 2015, 58(1): 36-45.)

The term *water flow pathway* in the mentioned paper of Yang et al. (2015) basically refers to the river network of a catchment. Although mentioned briefly in the beginning of our review as part of the discretisation process (page 4, line 6, termed *river segments*), the delineation of a river network was deliberately omitted from explicit discussion. It is merely implicitly contained as pre-processing procedure in any of the introduced concepts. Our focus, therefore, lies on the discretisation of terrestrial landscape elements where eventually the equations of the water balance shall be solved. We acknowledge, however, the importance of an adequate flow network delineation for hydrological modelling. Therefore, we shall see to briefly discuss this issue (and typical pre-processing tasks in general) in the revised version of our manuscript.

3) What are the basic units of hydrological simulation in different landscape
discretization approaches (different semi-distributed hydrological models)?

As we see it: the basic units of simulation in the different approaches are the spatial
units a specific approach is referring to, e.g., the grid cell in a raster-based model, the
HRUs in the HRU approach, the ASAs in the ASA approach. In more complex multi-
scale discretisation schemes (such as used for the WASA-SED model), however, a
basic unit cannot be clearly distinguished as several units are responsible to calculate
the water balance. In WASA-SED, for instance, that means: At SVC level infiltration,
evapotranspiration, and soil water movement are calculated; at TC level lateral runoff
re-distribution processes are simulated; at LU level groundwater is considered; at sub-
basin level the streamflow routing along a representative channel is assessed. We will
point this out in the revision and hope to have answered the question as desired.

4) In Section 2.3, it's hard to see what kinds of software is suitable for
landscape discretization?

Our goal in Sect. 2.3 was to give an overview over existing software and to identify
common limitations. We then tried to address these limitations in the development of a
new software package.

Section 3:

1) A flowchart is needed in description of lumpR. Figure 1 should be modi-
fied to contain both discretization procedure and major functions.

Actually, this is exactly what Fig. 1 does. It contains the package's functions (in italics)
along with a short description for every function. Furthermore, the arrows highlight

the flow of information, meaning the order of application of the functions (i.e., the procedure) to derive a complete hillslope-based landscape discretisation. We will try to clarify our intention (e.g., by using different arrow types or numbering the boxes) in the revision.

> 2) This section should be rearranged, for example, general workflow, major functions, additional tools.

Sects. 3.2 to 3.4 contain descriptions of the software's functions as corresponding to Fig. 1 (always outlined in italic letters). The order thus represents the typical processing sequence. However, as stated in the last point, we will seek for clarification during the revision of the manuscript.

> 3) How to define the topological relationships between the hillslopes and subcatchment?

The topological relationship between the hillslopes (or more precisely, the hillslope type representations inherent in the Landscape units) is derived in *lump_grass_post()* by intersecting the LUs with the subbasins. We will add this information to the respective section.

> 4) As shown in Figure 2, how to simulate the flow discharge in the river networks? It's also important to consider the spatial variability of precipitation inputs.

Figure 2 illustrates the conception of landscape discretisation in lumpR, i.e., the steps (i) to (iv) described in Sect. 3.2. It shall therefore mainly clarify the terms EHA, LU,

and TC, their relation to a subbasin, and what these terms represent in a real catchment. Although lumpR produces a river network as part of the pre-processing (function *lump_grass_prep()*), the actual simulation of discharge along the river network is not (and shall not be) part of the discretisation software but of the hydrological model which employs the output of lumpR. The pre-processing of the model forcing data (e.g., precipitation) is again not (and shall not be) part of lumpR.

5) This is the main part of this MS, more detailed information is required.

Apart from the changes announced above, we would kindly ask the reviewer for which specific aspects what additional information would be desirable.

Section 4:

1) Sensitivity analysis in this section is repeated in Section 5. I suggest move the sensitivity analysis to Section 5.

Section 4 describes the employed methodology of the SA and its results. Section 5 rather contains a discussion of the presented software package in general (Sect. 5.1) and the results of the sensitivity analysis for the case study (Sect. 5.2). As such, we prefer to leave the structure in its current state, as it better adheres to common conventions in separating methodology, results and discussion.

2) This section is very confusion. It's very hard to see how to implement the lumpR for application of WASA-SED model. Please rearrange this section.

Section 3 already describes the common application of lumpR. In contrast, Sect. 4 describes an advanced study that was enabled by the fully automatic massive replication of lumpR with different settings. We will clarify this in the introductory sentences of Sect. 4.
[Figure]

3) What are the inputs (climate forcing data) and outputs of WASA-SED model?

The input data of the model and their pre-processing are described in Sect. 4.3 of the manuscript. The model output used in this study is described in Sect. 4.4.2.

4) What are the major functions of WASA-SED model? Please show these functions by using the simulated results.

This manuscript shall serve as introduction into the software package lumpR as tool for hillslope-based landscape discretisation. The WASA-SED model has merely been used to exemplify its functionalities in a case study. A more detailed description of the WASA-SED model than given in Sect. 4.2 (and to some extent in the last paragraph of Sect. 2.2) is therefore not (and shall not be) part of this manuscript. For more detailed information, a reader should consider the given references of the model.

Other comments:

1) Yang et al. (2002) developed a hillslope-based hydrological model. This work should be cited. Yang, D, S Herath  K Musiake (2002), A Hillslope-based hydrological model using catchment area and width functions. Hydrological Sciences Journal, 47(1), 49- 65.

2) Some references should be added in Section 1, e.g. Yang et al., (2000), Comparison of different distributed hydrological models for characterization of catchment spatial variability. Hydrological Processes, 14, 403-416.

Thank you for giving some more examples of hillslope-based models. We shall consider the mentioned references in the revised version of our manuscript.

3) P4, L32-33: Here should emphasize that the models use HRUs face the difficulty of representing water flow pathways (for example the runoff from hillslopes and runoff along the river networks) appropriately.

We will add a line pursuing this concern.

4) P5, L15-20: The difficulty of determining the size of REW should be mentioned.

Thank you, we will further comment of the REW delineation in the revision of our manuscript.

5) P6, L15: A table is needed to summarize the existing software for model preprocessing

We shall add a table giving a summary of our review.

6) P12: A table is required to summarize the model parameters.

A summary of the most influential parameters of the lumpR package is given by Tab. 1. We will further clarify the reference to this table.

7) P19: The discharge hydrograph is required in addition to the reservoir water volume shown in Figure 4.

We omitted the discharge hydrograph as it is, due to the large number of zero flow events, much less informative than the reservoir volume shown in Fig. 4. We shall, however, consider putting a figure of the discharge hydrograph into the Appendix.

8) P 20: Use a table instead of Figure 5.

We use Fig. 5 to illustrate the properties of the distributions of the indices. This is important to, e.g., justify the use of the density-based sensitivity analysis method for skewed and multi-model distributions of the target variable. We therefore argue to keep Fig. 5 as it is instead of replacing it with a potentially less accentuating table which would only show numbers in a rather abstract way, instead of illustrating the eight distributions of 12250 values.

9) P 21: Show the results of sensitivity analysis using the hydrograph too

Actually, the results of the sensitivity analysis are based on the hydrograph (i.e., the time series of reservoir inflows), see Sect. 4.4.2. We will clarify this in the respective section

We hope to have addressed your concerns and answered your questions as expected.

Yours,

Tobias Pilz (on behalf of all co-authors)

---

## Author Comment (AC3) · 14 Mar 2017

Dear Reviewer,

thank you very much for your efforts on examining our manuscript. In the following we will answer your comments point-by-point.

General comments: Authors have developed an R package, lumpR, for watershed discretization in hillslope-based hydrological models. While developing an open source package for catchment discretization is the main goal of this manuscript, this paper does not introduce any new algorithm for hillslope delineation and semi-distributed hydrologic modeling that is superior to the existing approaches. As discussed in Section 2.3, series of
SHELL and MATLAB scripts are already available to process data for the
LUMP modeling approach. As a result, most of the paper reads like a user
manual for the R package.

This summary is correct regarding the novelty of single algorithms. However, the integrative approach realized in LumpR provides an innovative and automated combination of single approaches. We shall, therefore, point out that our software package contains more than a mere transformation of the existing SHELL and MATLAB packages. The existing code has been revised, checked and, where necessary, optimized. Furthermore, a number of additional processing steps and functionalities were added and the applicability was simplified. The presented sensitivity analysis was only possible by massive fully automated replicative application of lumpR over a (for a hillslope-based approach) relatively large scale ( 1000 $km^2$), which would have not been possible with any other existing software.

On the other hand, the sensitivity experiment described as the case study
in this paper is the strongest component of this paper. Therefore, I suggest
authors to expand on the sensitivity experiment by considering some of
the limitations they discuss as well as examining the impact of catchment
discretization on other components of the water budget.
How general are the findings from the sensitivity experiments? Do you expect to obtain similar results if you implement the approach in other basins?

Thank you for acknowledging our efforts on the sensitivity analysis. Generally, it needs to be stressed that the presented sensitivity analysis and the related conclusions are case-study and model specific.

It is therefore legitimate to ask for an extension of our analyses (also with respect to other components of the water budget). We would argue, however, that the main

objective of our manuscript is to introduce a newly compiled software package. Furthermore, it was our intention to do this along with a short review of existing algorithms and software for landscape discretisation and a case study which should contain at least a small scientific additional benefit. The latter we see in presenting a first and novel approach of sensitivity analysis of typical spatial discretisation parameters.

In our opinion, the extension of the paper by including more models, study sites and/or other water budget components lies beyond the scope of our manuscript, and also the GMD journal. The paper shall therefore rather serve as a stimulation of further research. In the revision we shall further comment on this in our manuscript as well.

Major comments:

1) Page 8-L10- Authors mention that they implemented the LUMP algorithm as the basis for development of the R package. However, it is not clear whether the semidistributed approach that is implemented in the LUMP model is superior to other discretization approaches in other semidistributed modeling approaches. What are the advantages of using the LUMP approach compared to other discritization approaches?

As we see it, hillslope-based discretization provides certain advantages (see a) below). LUMP and lumpR are just software tools to facilitate hillslope-based discretization. Their specific merits are recounted in b):

a. Regarding the advantages over other semidistributed approaches, a general feature of hillslope-based discretisation concepts is that it is especially useful in regions of heterogeneous runoff generation mechanisms (mentioned in page 5 L 11-13 of the manuscript and *Bronstert, 1999, 10.1002/(SICI)1099-1085(199901)13:1<21::AID-HYP702>3.0.CO;2-4*).

b. The main advantages of the LUMP algorithm over other hillslope-based discretisation algorithms is, first, its semi-automated nature allowing to automate and reproduce the discretisation process while retaining a certain degree of flexibility by allowing to integrate expert knowledge (supplemental information, specification of discretisation parameters, manual adjustments of intermediate results in GRASS). Second, contrary to other hillslope-based algorithms, via the multiscale discretisation it is applicable over large scales with relatively little effort (LUMP publication, *Francke et al., 2008, 10.1080/13658810701300873*). lumpR extends this merits by extending the options to fully-automated usage mode and optimized data handling, processing and storage.

We shall clarify these points in the revision of the manuscript.

2) Please switch Figure 1 with Figure 2 as it makes sense to have the discretization approach first and then the software structure.

We decided to put Fig. 2 to Sect. 3.2 after Fig. 1 as the latter demonstrates the general structure and workflow of the package whereas the former exemplifies the LUMP algorithm as specific part of the lumpR software package. For parity reasons, we chose not to extend too much on one particular approach in the review section. However, we acknowledge that Fig. 2 facilitates understanding the concept for readers unfamiliar with the this scheme. We added a reference to Fig. 2 in the review section.

3) Page 10-L10-Could you please explain Environmental Hillslope Areas?

Actually *environmental* is a typo in the manuscript and it should read *Elementary Hillslope Area*. It is the basic unit for calculation of a representative catena and, thus, one can think of it as a single specific hillslope. The EHA constitutes the basic element which the following steps of aggregation and generalization build upon. We thank

the Reviewer for pointing this out and acknowledge the insufficient description in the manuscript. We will improve the understanding in the revision.

> 4) In Section 3.2-It will be great if you can refer to different components of Figure 2 as you explain various functions.
>
> 5) Page 10-L20-What criteria are used to further subdivide each LU to terrain components?

We will try to clarify Sect. 3 as a whole by better referring to the presented graphics and presenting more details of the algorithms. To subdivide the LU into TC a parameter has to be specified giving the number of TC to be generated. The partition is done by evaluating the derived LU (i.e., the averaged representative toposequence) properties and employing a minimisation of variances approach (described in the LUMP paper of *Francke et al., 2008, 10.1080/13658810701300873*).

> 6) Figure 4- It seems reservoir volume is always overestimated compared to observations for various sensitivity runs. Can authors explain the reason for this behavior?

There are several potential reasons for this observation. As the most likely can be considered: (i) uncertainties in the reservoir parameterisation in the model (e.g., we use a static parameterisation of reservoir abstractions which are, in reality, dynamic; furthermore, sedimentation in the model, i.e., loss of reservoir volume, is in our case not considered); (ii) uncertainties in the observations (e.g., due to deficiencies of the rating curve); (iii) rainfall input uncertainty (as is also discussed at page 24 L 4 ff.); (iv) model parameterisation uncertainty (the model has been run with standard parameterisation for the area without further calibration). We shall expand Sect. 5.2 accordingly (or re-structure Sect. 5).

7) Figure 6- It is not clear how general is the results of this figure. Do authors predict that similar sensitivity pattern will be obtained if a different hydrologic model is used?

Actually, this is a question we have in mind as well. Apart from speculations we cannot give an answer yet. Generally, it needs to be pointed out that the presented sensitivity analysis is case-study and model specific. Please also respect our general comments further above on that issue (the second point of answers).

8) Additional information about soil and land-cover parameters are required and how discretization approach impacted parameter selection for each hill-slope.

The different discretisations did not affect soil nor land-cover parameters. They merely modify the fractions of soil and vegetation types that are assigned to the spatial units. We hope this answers your question. We will explicitly point this out in our revision.

We hope the stated concerns and questions have been addressed sufficiently and as expected.

Yours,

Tobias Pilz (on behalf of all co-authors)

---

## Author Response (AR1)

**Author's response**

We would like to thank the two anonymous reviewers and Mrs. Kerkweg for their time spent on examining our manuscript and for their helpful suggestions and comments. We belief that we could improve the readability of the paper and the presentation of our methods and key messages during the revision. A marked-up version of the revised manuscript is appended at the end of this document. In the following we list the Referee comments (indented and in italics) together with our responses. The latter are based on our responses during the public discussion process (see the Author Comments at the discussion website). We hope to have satisfactorily addressed all raised concerns.

**SC1 by A. Kerkweg**

> *Dear authors,*
>
> *in my role as Executive editor of GMD, I would like to bring to your attention our Editorial version 1.1:*
>
> *http://www.geosci-model-dev.net/8/3487/2015/gmd-8-3487-2015.html*
>
> *This highlights some requirements of papers published in GMD, which is also available on the GMD website in the 'Manuscript Types' section:*
>
> *http://www.geoscientific-model-development.net/submission/manuscript_types.html*
>
> *In particular, please note that for your paper, the following requirement has not been met in the Discussions paper:*
>
> > *"The main paper must give the model name and version number (or other unique identifier) in the title."*
>
> *Please add a version number for lumpR in the title upon your revised submission to GMD.*
>
> *Yours,*
>
> *Astrid Kerkweg*

We added the version number to the title.

**RC1 by Anonymous Referee #1**

> *General comments: This manuscript introduced a computer software lumpR that serves as pre-processing tool for the hydrological model WASA-SED. This manuscript is more like a technical document, it needs to emphasize the key features and major functions of this software. It also declared that "the first objective of this paper is to provide an overview of existing landscape discretisation algorithms and software", however this review need a major revision.*

We revised the manuscript in order to improve readability and understanding with special focus on Sects. 1 (Introduction) and 2 (the Review).

> *There are many literature reviews in each sections. Please move these literature reviews into Section 1 or/and Section 2.*

We agree that literature reviews should be rather concentrated. That is why we concentrated our literature reviews in the manuscript in Sects. 1 and 2. The cited literature in the other sections is not part of a review, instead to refer to the specific algorithms implemented in the software (Sects. 3.2 and 3.3), introduce the study site and the employed hydrological model (Sects. 4.1 and 4.2, respectively), citation of data sources (Sect. 4.3), to explain the method used for sensitivity analysis of the software's parameters (Sect. 4.4), and to discuss the obtained results along with other studies (Sect. 5). We would therefore kindly ask for a more specific description of this concern.

*Section 1: Introduction*

*1) Classification of the hydrological modelling approaches into three types (fully distributed approach, lumped approach and semi-distributed approach) is often questioned. For example, many large scale hydrological model uses grid cells for the discretisation of the landscape, but the size of the grid cell is very large, so they also employ sub-grid parameterization schemes. What type do these hydrological models belong to?*

We agree that the classification approach of hydrological models into the mentioned three categories can be ambiguous and is not always well-defined. We adapted the introduction in a way pointing out possible ambiguities in the conception of the classification in fully distributed, lumped approach and semi-distributed discretisations. We explicitly referred to sub-grid parameterisation as another strategy of increasing the degree of detail in models without increasing the spatial resolution (see second paragraph of the revised introduction).

*2) The logic between the literature reviews in Section 1 and Section 2 is not clear*

In the introduction (Sect. 1), a classification of landscape discretisation concepts in hydrological modelling into fully distributed, semi-distributed, and lumped approaches is made while briefly explaining advantages and limitations of each concept. In the review (Sect. 2), however, the focus solely lies on semi-distributed approaches. In the revised version of the manuscript, tried to better point out that objective (third paragraph of introduction).

*3) The logic between the first objective and the second objective is not clear. Before the the overview of existing landscape discretisation algorithms, authors already determined to use hillslope-based approach.*

The first objective of the paper is to present existing landscape discretisation algorithms and software solutions along with their limitations. We introduced, as second objective, a new software picking up the identified limitations. Our intended logic between the two was to demonstrate the impetus for presenting yet a further software. However, we hope to have clarified that point in the revised manuscript.

*Section 2: Review of landscape discretisation in hydrological modelling*

*1) This section should focus on the landscape representation, not only the landscape discretization.*

Thank you, adapted the title of Sect. 2. Indeed, the review is already focussing on landscape representation as a whole rather than the mere discretisation procedure.

*2) Section 2.1 presents common knowledge about the DEM. What is relationship with section 2.2?*

We acknowledge that Sect. 2.1 currently stands a bit apart from Sect. 2.2. Our intention was to first give a short overview over different topography representation approaches before discussing landscape discretisation as the former are the source of data for any landscape analysis and discretisation. By extending the introductory sentences of Sect. 2, we hope to have established a more profound relation between the two subsections.

*2) To model the catchment hydrology, water flow pathway is a key issue in landscape discretization/representation. This section lacks of representation of catchment water flow pathway. (Reference: Yang D, Gao B, Jiao Y, et al. A distributed scheme developed for eco-hydrological modeling in the upper Heihe River. Science China Earth Sciences, 2015, 58(1): 36-45.)*

The term *water flow pathway* in the mentioned paper of Yang et al. (2015) basically refers to the river network of a catchment. The delineation of a river network was deliberately omitted from explicit discussion. It is merely implicitly contained as preprocessing procedure in any of the introduced concepts. Our focus, therefore, lies on the discretisation of terrestrial landscape elements where eventually the equations of the water balance shall be solved. We acknowledge, however, the importance of an adequate flow network delineation for hydrological modelling. Therefore, we added this issue (and typical pre-processing tasks in general) in the revised version of our manuscript (introductory sentences of Sect. 2).

*3) What are the basic units of hydrological simulation in different landscape discretization approaches (different semi-distributed hydrological models)?*

As we see it: the basic units of simulation in the different approaches are the spatial units a specific approach is referring to, e.g., the grid cell in a raster-based model, the HRUs in the HRU approach, the ASAs in the ASA approach. In more complex multi-scale discretisation schemes (such as used for the WASA-SED model), however, a basic unit cannot be clearly distinguished as several units are responsible to calculate the water balance. In WASA-SED, for instance, that means: At SVC level infiltration, evapotranspiration, and soil water movement are calculated; at TC level lateral runoff re-distribution processes are simulated; at LU level groundwater is considered; at subbasin level the streamflow routing along a representative channel is assessed. We hope to have answered the question as desired and clarified that point in the revised manuscript.

*4) In Section 2.3, it's hard to see what kinds of software is suitable for landscape discretization?*

Our goal in Sect. 2.3 was to give an overview over existing software and to identify common limitations. We then tried to address these limitations in the development of a new software package.

*Section 3:*

*1) A flowchart is needed in description of lumpR. Figure 1 should be modified to contain both discretization procedure and major functions.*

Actually, this is exactly what Fig. 1 does. It contains the package's functions (in italics) along with a short description for every function. Furthermore, the arrows highlight the flow of information, meaning the order of application of the functions (i.e., the procedure) to derive a complete hillslope-based landscape discretisation. We further added numbers to Fig. 1 and adapted Sect. 3 in order to make the workflow better understandable.

*2) This section should be rearranged, for example, general workflow, major functions, additional tools.*

Sects. 3.2 to 3.4 contain descriptions of the software's functions as corresponding to Fig. 1 (always outlined in italic letters). The order thus represents the typical processing sequence. However, as stated in the last point, we hope to have clarified the workflow.

*3) How to define the topological relationships between the hillslopes and subcatchment?*

The topological relationship between the hillslopes (or more precisely, the hillslope type representations inherent in the Landscape units) is derived in *lump_grass_post()* by intersecting the LUs with the subbasins. We added this information to Sect. 3.2.

*4) As shown in Figure 2, how to simulate the flow discharge in the river networks? It's also important to consider the spatial variability of precipitation inputs.*

Figure 2 illustrates the conception of landscape discretisation in lumpR, i.e., the steps (i) to (iv) described in Sect. 3.2. It shall therefore mainly clarify the terms EHA, LU, and TC, their relation to a subbasin, and what these terms represent in a real catchment. Although lumpR produces a river network as part of the pre-processing (function *lump_grass_prep()*), the actual simulation of discharge along the river network is not (and shall not be) part of the discretisation software but of the hydrological model which employs the output of lumpR. The pre-processing of the model forcing data (e.g., precipitation) is again not (and shall not be) part of lumpR.

*5) This is the main part of this MS, more detailed information is required.*

We hope to have addressed all concerns within the aforementioned points and that the suggestions have been implemented in the most adequate manner.

*Section 4:*

*1) Sensitivity analysis in this section is repeated in Section 5. I suggest move the sensitivity analysis to Section 5.*

Section 4 describes the employed methodology of the SA and its results. Section 5 rather contains a discussion of the presented software package in general (Sect. 5.1) and the results of the sensitivity analysis for the case study (Sect. 5.2). As such, we prefer to leave the structure in its current state, as it better adheres to common conventions in separating methodology, results and discussion.

*2) This section is very confusion. It's very hard to see how to implement the lumpR for application of WASA-SED model. Please rearrange this section.*

Section 3 already describes the common application of lumpR. In contrast, Sect. 4 describes an advanced study that was enabled by the fully automatic massive replication of lumpR with different settings. We believe this is covered by the introductory sentences of Sect. 4.

*3) What are the inputs (climate forcing data) and outputs of WASA-SED model?*

The input data of the model and their pre-processing are described in Sect. 4.3 of the manuscript. The model output used in this study is described in Sect. 4.4.2.

*4) What are the major functions of WASA-SED model? Please show these functions by using the simulated results.*

This manuscript shall serve as introduction into the software package lumpR as tool for hillslope-based landscape discretisation. The WASA-SED model has merely been used to exemplify its functionalities in a case study. A more detailed description of the WASA-SED model than given in Sect. 4.2 (and to some extent in the last paragraph of Sect. 2.2) is therefore not (and shall not be) part of this manuscript. For more detailed information, a reader should consider the given references of the model.

*Other comments:*

*1) Yang et al. (2002) developed a hillslope-based hydrological model. This work should be cited. Yang, D, S Herath & K Musiake (2002), A Hillslope-based hydrological model using catchment area and width functions. Hydrological Sciences Journal, 47(1), 49- 65.*

*2) Some references should be added in Section 1, e.g. Yang et al., (2000), Comparison of different distributed hydrological models for characterization of catchment spatial variability. Hydrological Processes, 14, 403-416.*

Thank you for giving some more examples of hillslope-based models. We considered the mentioned references in the revised version of our manuscript.

*3) P4, L32-33: Here should emphasize that the models use HRUs face the difficulty of representing water flow pathways (for example the runoff from hillslopes and runoff along the river networks) appropriately.*

To bring out this concern, we added the following line to Sect. 2.2: *Instead of a direct representation of water flow pathways, generated runoff is typically summed over all HRUs of a watershed and routed along a representative channel element.*

*4) P5, L15-20: The difficulty of determining the size of REW should be mentioned.*

Added the following sentence to Sect. 2.2 when describing functional units and the REW approach: *However, it still is challenge how to define the size of averaging volumes and the closure relationships of boundary fluxes (Beven, 2006).*

*5) P6, L15: A table is needed to summarize the existing software for model preprocessing*

In the revision, we added Tab. 1 to give an overview over landscape discretisation approach and software examples.

*6) P12: A table is required to summarize the model parameters.*

A summary of the most influential parameters of the lumpR package is given by Tab. 1. We added a reference to this table in Sect. 3.2.

*7) P19: The discharge hydrograph is required in addition to the reservoir water volume shown in Figure 4.*

We omitted the discharge hydrograph as it is, due to the large number of zero flow events, much less informative than the reservoir volume shown in Fig. 4. We added a sentence pointing this out at the beginning of Sect. 4.5.

*8) P 20: Use a table instead of Figure 5.*

We use Fig. 5 to illustrate the properties of the distributions of the indices. This is important to, e.g., justify the use of the density-based sensitivity analysis method for skewed and multi-model distributions of the target variable. We therefore argue to keep Fig. 5 as it is instead of replacing it with a potentially less accentuating table which would only show numbers in a rather abstract way, instead of illustrating the eight distributions of 12250 values.

*9) P 21: Show the results of sensitivity analysis using the hydrograph too*

Actually, the results of the sensitivity analysis are based on the hydrograph (i.e., the time series of reservoir inflows), see Sect. 4.4.2. To avoid confusion, we clarified Sect. 4.5 (note also our answer above).

**RC2 by Anonymous Referee #2**

*General comments: Authors have developed an R package, lumpR, for watershed discretization in hillslope-based hydrological models. While developing an open source package for catchment discretization is the main goal of this manuscript, this paper does not introduce any new algorithm for hillslope delineation and semi-distributed hydrologic modeling that is superior to the existing approaches. As discussed in Section 2.3, series of SHELL and MATLAB scripts are already available to process data for the LUMP modeling approach. As a result, most of the paper reads like a user manual for the R package.*

This summary is correct regarding the novelty of single algorithms. However, the integrative approach realized in LumpR provides an innovative and automated combination of single approaches. We shall, therefore, point out that our software package contains more than a mere transformation of the existing SHELL and MATLAB packages. The existing code has been revised, checked and, where necessary, optimized. Furthermore, a number of additional processing steps and functionalities were added and the applicability was simplified. The presented sensitivity analysis was only possible by massive fully automated replicative application of lumpR over a (for a hillslope-based approach) relatively large scale ( 1000 km$^2$), which would have not been possible with any other existing software.

*On the other hand, the sensitivity experiment described as the case study in this paper is the strongest component of this paper. Therefore, I suggest authors to expand on the sensitivity experiment by considering some of the limitations they discuss as well as examining the impact of catchment discretization on other components of the water budget.*

*How general are the findings from the sensitivity experiments? Do you expect to obtain similar results if you implement the approach in other basins?*

Thank you for acknowledging our efforts on the sensitivity analysis. Generally, it needs to be stressed that the presented sensitivity analysis and the related conclusions are case-study and model specific.
It is therefore legitimate to ask for an extension of our analyses (also with respect to other components of the water budget). We would argue, however, that the main objective of our manuscript is to introduce a newly compiled software package. Furthermore, it was our intention to do this along with a short review of existing algorithms and software for landscape

discretisation and a case study which should contain at least a small scientific additional benefit. The latter we see in presenting a first and novel approach of sensitivity analysis of typical spatial discretisation parameters.

In our opinion, the extension of the paper by including more models, study sites and/or other water budget components lies beyond the scope of our manuscript, and also the GMD journal. The paper shall therefore rather serve as a stimulation of further research. We added some lines pursuing these statements at the ends of Sects. 5 and 6, respectively.

*Major comments:*

*1) Page 8-L10- Authors mention that they implemented the LUMP algorithm as the basis for development of the R package. However, it is not clear whether the semidistributed approach that is implemented in the LUMP model is superior to other discretization approaches in other semi-distributed modeling approaches. What are the advantages of using the LUMP approach compared to other discritization approaches?*

As we see it, hillslope-based discretization provides certain advantages (see a) below). LUMP and lumpR are just software tools to facilitate hillslope-based discretization. Their specific merits are recounted in b):

a. Regarding the advantages over other semidistributed approaches, a general feature of hillslope-based discretisation concepts is that it is especially useful in regions of heterogeneous runoff generation mechanisms (mentioned in page 5 L 11-13 of the manuscript and *Bronstert, 1999, 10.1002/(SICI)1099-1085(199901)13:1<21::AID-HYP702>3.0.CO;2-4*).

b. The main advantages of the LUMP algorithm over other hillslope-based discretisation algorithms is, first, its semi-automated nature allowing to automate and reproduce the discretisation process while retaining a certain degree of flexibility by allowing to integrate expert knowledge (supplemental information, specification of discretisation parameters, manual adjustments of intermediate results in GRASS). Second, contrary to other hillslope-based algorithms, via the multi-scale discretisation it is applicable over large scales with relatively little effort (LUMP publication, *Francke et al., 2008, 10.1080/13658810701300873*). lumpR extends this merits by extending the options to fully-automated usage mode and optimized data handling, processing and storage.

We extended the second to last paragraph of Sect. 2.3 to make it clearer.

*2) Please switch Figure 1 with Figure 2 as it makes sense to have the discretization approach first and then the software structure.*

We decided to put Fig. 2 to Sect. 3.2 after Fig. 1 as the latter demonstrates the general structure and workflow of the package whereas the former exemplifies the LUMP algorithm as specific part of the lumpR software package. For parity reasons, we chose not to extend too much on one particular approach in the review section. However, we acknowledge that Fig. 2 facilitates understanding the concept for readers unfamiliar with the this scheme. We added a reference to Fig. 2 in the second to last paragraph of Sect. 2.3.

*3) Page 10-L10-Could you please explain Environmental Hillslope Areas?*

Actually *environmental* is a typo in the manuscript and it should read *Elementary Hillslope Area*. It is the basic unit for calculation of a representative catena and, thus, one can think of it as a single specific hillslope. The EHA constitutes the basic element which the following steps of aggregation and generalization build upon. We thank the Reviewer for pointing this out and acknowledge the insufficient description in the manuscript. In order to improve understanding, we updated Sect. 3.2 accordingly.

*4) In Section 3.2-It will be great if you can refer to different components of Figure 2 as you explain various functions.*

*5) Page 10-L20-What criteria are used to further subdivide each LU to terrain components?*

We updated Sect. 3.2, now referring to Fig. 2 after each processing step, and explaining the derivation of TCs in a more detailed way.

*6) Figure 4- It seems reservoir volume is always overestimated compared to observations for various sensitivity runs. Can authors explain the reason for this behavior?*

We expanded Sect. 5.2 discussing that issue.

*7) Figure 6- It is not clear how general is the results of this figure. Do authors predict that similar sensitivity pattern will be obtained if a different hydrologic model is used?*

Actually, this is a question we have in mind as well. Apart from speculations we cannot give an answer yet. Generally, it needs to be pointed out that the presented sensitivity analysis is case-study and model specific. Please also respect our general comments further above on that issue (the second point of answers).

*8) Additional information about soil and land-cover parameters are required and how discretization approach impacted parameter selection for each hillslope.*

[revised manuscript text omitted]